

# A dedicated flask sampling strategy developed for ICOS stations based on CO₂ and CO measurements and STILT footprint modelling

Ingeborg Levin[1], Ute Karstens[2], Markus Eritt[3], Fabian Maier[1], Sabrina Arnold[4], Daniel Rzesanke[3], Samuel Hammer[1], Michel Ramonet[5], Gabriela Vítková[6], Sebastien Conil[7], Michal Heliasz[8], Dagmar Kubistin[4], and Matthias Lindauer[4]

[1]Institut für Umweltphysik, Heidelberg University, INF 229, 69120 Heidelberg, Germany
[2]ICOS Carbon Portal, Lund University, Geocentrum II, Sölvegatan 12, 22362 Lund, Sweden
[3]Max Planck Institute for Biogeochemistry, ICOS Flask- und Kalibrierlabor, Kahlaische Strasse 4, 07745 Jena, Germany
[4]Meteorologisches Observatorium Hohenpeißenberg, Deutscher Wetterdienst, Albin-Schwaiger-Weg 10, 82383 Hohenpeißenberg, Germany
[5]Laboratoire des Sciences du Climat et de l'Environnement (LSCE), IPSL, CEA-CNRS-UVSQ, Université Paris-Saclay, Orme des Merisiers, Bât.714, 91191 Gif-sur-Yvette, France
[6]Global Change Research Institute of the Czech Academy of Sciences, Bělidla 986/4°, 603 00 Brno, Czech Republic
[7]DRD/OPE, Andra, RD960 - BP9, 55290 Bure, France
[8]Centre for Environmental and Climate Research, Lund University, Sölvegatan 37, 223 62 Lund, Sweden

*Correspondence to*: Ingeborg Levin (Ingeborg.Levin@iup.uni-heidelberg.de)





**Abstract.** In situ $CO_2$ and CO measurements from five atmospheric ICOS (Integrated Carbon Observation System) stations have been analysed together with footprint model runs from the regional transport model STILT, to develop a dedicated strategy for flask sampling with an automated sampler. Flask sampling in ICOS has three different purposes: 1) Provide an independent quality control for in situ observations, 2) provide representative information on atmospheric components

currently not monitored in situ at the stations, 3) collect samples for $^{14}CO_2$ analysis that are significantly influenced by fossil fuel $CO_2$ (ff$CO_2$) emission areas. Based on the existing data and experimental results obtained at the Heidelberg pilot station with a prototype flask sampler, we suggest that single flask samples should be collected regularly every third day around noon/afternoon from the highest level of a tower station. Air samples shall be collected over one hour with equal temporal weighting to obtain a true hourly mean. At all stations studied, more than 50 % of flasks to be collected around mid-day will

likely be sampled during low ambient variability (<0.5 ppm standard deviation of one-minute values). Based on a first application at the Hohenpeißenberg ICOS site, such flask data are principally suitable to detect $CO_2$ concentration biases larger than 0.1 ppm with a one-sigma confidence level between flask and in situ observations from only 5 flask comparisons. In order to have a maximum chance to also sample ff$CO_2$ emission areas, additional flasks need to be collected on all other days in the afternoon. Using the continuous in situ CO observations, the CO deviation from an estimated background value must be

determined the day after each flask sampling and, depending on this offset, an automated decision must be made if a flask shall be retained for $^{14}CO_2$ analysis. It turned out that, based on existing data, ff$CO_2$ events of more than 4-5 ppm will be very rare  at all stations studied, particularly in summer. During the other seasons, events could be collected more frequently. The strategy developed in this project is currently being implemented at the ICOS stations.

## 1 Introduction

Since the pioneering work by Charles David Keeling who, already in the 1950s, has started monitoring with in situ instrumentation continuous atmospheric carbon dioxide concentration at South Pole and Mauna Loa (Brown and Keeling, 1965), global coverage of continuous greenhouse gas (GHG) observations has considerably improved (https://ds.data.jma.go.jp/gmd/wdcgg/). However, there still exist large observational gaps in remote regions of the globe, which have partly been filled by regular flask sampling with subsequent GHG analysis in central laboratories. In the marine

realm, if frequently conducted under certain conditions, data from flask sampling are often representative for monitoring the large-scale distribution of GHGs in the atmosphere and, respectively, for estimating large-scale flux distributions by inverse modelling.

In the last decades, observational networks have been extended to the continents in order to closely monitor GHG

concentrations and quantify terrestrial GHG fluxes. These, however, are more heterogeneous, temporally variable and often less well represented by models than it is the case with modelled ocean fluxes (Friedlingstein et al., 2019). Terrestrial biospheric fluxes are prone to (regional) climatic variability and changes, and only continental observations provide the gateway for process understanding. Besides monitoring the terrestrial biosphere, measurements over continents are also conducted to observe man-made emissions, in particular from fossil fuel burning and agriculture. Due to their proximity to

these highly variable sources and sinks, measurements over continents are best conducted continuously with in situ instrumentation at high temporal resolution, in order to cover the variability and to fully represent the entire footprint of a station (e.g. Andrews et al., 2014). However, not all atmospheric trace components to be included in continental top-down GHGs budgeting can yet be precisely measured in situ at remote stations. The most prominent example is radiocarbon ($^{14}C$) in atmospheric $CO_2$, a quantitative tracer to separate the fossil from the biospheric component in recently emitted $CO_2$ from

continental sources (e.g. Levin et al., 2003). Note that in the industrialised and highly populated areas of mid latitudes of the



Northern Hemisphere, i.e. in North America, Eastern Asia or Europe, atmospheric signals from the biosphere and from fossil fuel sources are of same order (see Sect. 4.3.1). To correctly interpret absolute $CO_2$ concentration variations in terms of source/sink attribution, separation of the fossil from the biogenic $CO_2$ signal is, therefore, mandatory. Precise $^{14}CO_2$ measurements are, however, currently only possible in dedicated laboratories and on discrete samples.


In Europe the Integrated Carbon Observation System Research Infrastructure (ICOS RI) (https://www.icos-ri.eu/icos-research-infrastructure) has been established to monitor GHGs concentrations and fluxes in the atmosphere, in various ecosystems and over the neighbouring ocean basins. ICOS atmosphere has set up a pan-European network of preferentially tall tower stations located at least 50 km away from industrialised and highly populated areas. The primary purpose is to monitor biogenic sources

and sinks in Europe and their behaviour under changing climatic conditions. In addition to continuous $CO_2$, $CH_4$ and $CO$ observations, a subset of stations (Class 1 stations) perform two-week integrated sampling of $CO_2$ for $^{14}C$ analysis. Class 1 stations are additionally equipped with an automated flask sampler, dedicated to three major objectives. Firstly, the collected flasks shall provide an independent quality control (QC) for the continuous in situ measurements of $CO_2$, $CH_4$, $CO$ and further species mole fractions. Secondly, flasks shall be collected for analysis of additional trace components not measured in situ at

the stations, and finally flasks with a potentially elevated fossil fuel $CO_2$ component originating from anthropogenic sources in the footprint of the stations shall be analysed for $^{14}CO_2$.

Dedicated sampling strategies had to be developed for ICOS, which best meet these three objectives, and which can be accomplished in the framework of the infrastructure and its available capabilities and resources. This includes technical

constraints at the stations but also analysis capacity at the ICOS Central Analytical Laboratories, which are analysing all flask samples in ICOS. The ICOS flask sampling strategy might change in future, e.g. when real-time GHG or footprint prediction tools become available.

In the current paper, we first give an introduction to the current ICOS atmospheric station network, and then present a strategy

how to collect the flask samples for ICOS in a simple and cost-effective way. The sampling strategies have been developed based on footprint model simulations with the regional transport model STILT (Lin et al., 2003), that was implemented at the ICOS Carbon Portal (https://www.icos-cp.eu/about-stilt) for ICOS station PIs and data users. First tests to develop a strategy for the quality control objective were performed at the ICOS pilot station in Heidelberg, where ICOS instrumentation and a prototype of the ICOS flask sampler have been installed, as well as at the Hohenpeißenberg station. The strategy was further

tested for its feasibility based on the first years of continuous ICOS $CO_2$ and $CO$ observations available at the ICOS Carbon Portal (ICOS RI, 2019).

## 2 The atmospheric component of the ICOS Research Infrastructure

### 2.1 The atmospheric station network and its Central Facilities

The ICOS atmospheric station network currently consists of 23 official stations (with 14 stations more to come), located in 12

countries and covering Europe from Scandinavia to Italy and from Great Britain to Czech Republic (see Fig. 1). The preferred station type are tall tower sites, allowing vertical profile sampling at a minimum of three height levels up to at least 100m a.g.l. Tall tower stations cover footprints of several 10 to 100 km distance from the sites (Gloor et al, 2001; Gerbig et al., 2006). Although their representation in state-of-the-art regional atmospheric transport models is more difficult than in the case of tower observations, due to their often long history of GHG measurements also a number of mountain and coastal stations are

part of the ICOS network. However, the flask sampling strategy developed here was designed specifically for the standard ICOS tall tower stations.



All ICOS atmosphere stations are equipped with commercially available instruments measuring continuously at high temporal resolution $CO_2$, $CH_4$ and CO. Instruments are tested at the Atmospheric Thematic Centre (ATC), an ICOS Central Facility

hosted by LSCE in Gif-sur-Yvette, France, before they are installed at the sites (Yver Kwok et al., 2015). The calibration gases for the in situ measurements are prepared and calibrated at the Flask and Calibration Laboratory (FCL), which has been established at the Max Planck Institute for Biogeochemistry in Jena, Germany, as part of the ICOS Central Analytical Laboratories (CAL). This procedure guarantees best possible compatibility of observations within the ICOS atmospheric network and maintaining the link to the internationally accepted WMO calibration scales. In addition, FCL analyses the flasks

with the focus on QC and additional species. Precise $^{14}CO_2$ analysis of integrated samples and selected flasks is conducted in the second part of ICOS CAL at Heidelberg University, Institute of Environmental Physics, at the Karl Otto Münnich Central Radiocarbon Laboratory (CRL).

All raw data (level-0) are automatically transferred, on a daily basis, from the measurement sites to the ATC where they are

converted to calibrated (level-1) concentration values (Hazan et al., 2016), based on regular on-site calibrations and FCL-assigned calibration values. For ongoing automatic data quality assurance of all measurements, the ATC has developed automatic procedures. Further software tools are made available by the ATC for mandatory validation of all raw data by the station PIs. These quality-assessed data form the basis of the hourly mean concentrations, which are finally released as level-2 data and made available to the user community at the ICOS Carbon Portal, hosted by Lund University, Sweden. For the

latest data release see ICOS RI (2019).

Two station types are currently implemented in the ICOS atmospheric station network, Class 1 and Class 2. Class 1 stations are equipped with the complete instrumentation including integrated $^{14}CO_2$ and flask sampling. Class 2 stations perform only in situ continuous measurements of $CO_2$, $CH_4$ and CO (currently not mandatory), but with the same instrumentation and

demand on data quality as for Class 1 stations. A detailed description of the specifications of instrumentation is given in the ICOS Atmospheric Station Specification document (https://icos-atc.lsce.ipsl.fr/filebrowser/download/69422), which is regularly updated. To become official part of the ICOS atmospheric station network, stations have to undergo a two-step labelling process, which shall warrant their conformance with the ICOS station specifications, including smooth data transfer to the ATC as well as meeting ICOS data quality requirements.

**2.2 Description of selected ICOS stations**

For developing and testing our flask sampling strategy we selected five ICOS Class 1 tall tower stations in four different countries. A short description of these stations is given in the following.

**Hyltemossa** (HTM) is located few kilometres south of Perstorp, in north western Skåne, Sweden (56.098° N, 13.418° E, 115

m a.s.l.). It hosts a combined atmospheric and ecosystem station, labelled respectively as Class 1 and Class 2 site in its respective networks. The site was established in 2014 in a 30 year old managed Norway spruce forest. Further than 600 m from the tower there is a mosaic consisting of forests, clear cuts and farm fields. Within the radius of 100 km, the elevation changes between 0–200 m a.s.l., while in the near vicinity of the tower elevation gently changes only by 35 m. In the larger footprint, the site is surrounded by cities, i.e. to the north Halmstad (70 km, 58 000 inhabitants), to the east Kristianstad (45

km, 36 000 inhabitants), to the south-west Lund (45 km, 111 000 inhabitants), Malmö (60km, 318 000 inhabitants) and Copenhagen, DK (70 km, 1 990 000 inhabitants) and to the west Helsingborg (45 km, 124 000 inhabitants) and Helsingør, DK (55 km, 61 000 inhabitants). The station is equipped with a Picarro G2401 Cavity Ring-Down Spectroscopy (CRDS) gas analyser measuring $CO_2$, $CH_4$ and CO. Air inlets are located at 30 m, 70 m and 150 m a.g.l. Air is being sampled for 5 min



from each level where data for the first minute after switching are discarded. Sampling lines have installed eight litre mixing
volumes continuously flushed with a flow rate of 1.6 L min$^{-1}$, resulting in a residence time of 215 sec in each line. In addition
at the height of each air inlet, air temperature, relative humidity as well as wind speed and direction are being measured.

The ICOS tall tower station **Gartow** (GAT, 53.066° N, 11.443° E, 70 m a.s.l.) is situated in the easternmost tip of Lower
Saxony, Germany, close to the river Elbe, approximately in the middle between Hamburg and Berlin. The surrounding area is
very flat with elevations ranging from less than 9 m a.s.l. (Elbe valley) up to 124 m a.s.l. (hill "Hoher Mechtin" – 35 km west
of GAT). The land use in this area is dominated by forests and agricultural fields. The station is hosted by a lattice television
tower operated and managed by the DFMG (Deutsche Funkturm GmbH). The closest cities are Schwerin (65 km north of the
station, ca. 100 000 inhabitants), Wolfsburg (80 km south of the station, ca. 120 000 inhabitants), and Lüneburg (70 km
northwest of the station, ca. 70 000 inhabitants). Air inlets are at 30 m, 60 m, 132 m, 216 m and 341 m. A Picarro G2301
Cavity Ring-Down Spectroscopy (CRDS) gas analyser for measuring $CO_2$, $CH_4$ and CO, and since beginning of 2019, a Los
Gatos Research 913-0015 (EP) Off-axis integrated cavity output spectroscopy (OA-ICOS) analyser measuring CO and $N_2O$
are installed in a container next to the tower. Air is being sampled for 5 min from each level where data for the first minute
after switching are discarded. All inlet lines are continuously flushed with approx. 5 L min$^{-1}$. Meteorological sensors for air
temperature, relative humidity, wind speed and direction are installed at every sampling height. For historical reasons, Gartow
modelling was conducted for 344 m a.g.l. (not for the highest sampling level, at 341 m); this difference between measured and
modelled level is, however, not relevant for the comparisons presented in the context of this study.

Station **Křešín** u Pacova (KRE) (49.572° N, 15.080° E, 534 m a.s.l.) is located in the central Czech Republic, about 100km
south-east of Prague. The site was established in 2013 close to the Košetice Observatory – station of the Czech
Hydrometeorological Institute with 30 years of practice in meteorology as well as air quality monitoring. Today, these two
stations form the National Atmospheric Observatory. Since the site is designed as a background station, the area is not
significantly influenced by human activity. The tower is surrounded by fields and, at a greater distance, by forests and small
villages (the closest in 1 km distance). There is a highway running north-east of the tower at approx. 6 km distance, however,
the wind frequencies from north and east are 9 and 5 %, respectively. The closest towns, Pelhřimov, Vlašim and Humpolec,
with 10 to 17 thousand inhabitants, are located approx. 20 km away from the station. As for industrial activity, a small wood-
processing company is located 20 km to the west (which is the prevailing wind direction). Town Havlíčkův Brod with ca.
20 000 inhabitants, is located about 30 km from the site, larger towns (up to 50 000 inhabitants) are about 40 km away (Jihlava,
Tábor). Farther there are only towns with populations of max. 35 000 inhabitants, except for Prague (80 km, one million
inhabitants), Pardubice (80 km, 90 000 inhabitants) and České Budějovice (90 km, 90 000 inhabitants). The terrain around the
tower is relatively flat within a few kilometres distance, with only small hills around. The Czech-Moravian Highlands, where
the site is located, have an average altitude of 500-600 m a.s.l. with rare spots of 800 m a.s.l. The highest hills, Javořice (837
m a.s.l.) and Devět skal (836 m a.s.l.) are located 43 km and 69 km away. The station is equipped with the ICOS atmosphere
recommended instrumentation for $CO_2$ and $CH_4$ (a Picarro G2301 CRDS) and for $N_2O$ and CO (Los Gatos Research 913-0015
(EP)). The air is sampled at levels 10 m, 50 m, 125 m and 250 m of the tower. Sampling period is 10 minutes per height where
the highest level is sampled in between all other levels. This results in a complete vertical profile measured within one hour
with preference of the 250 m level. After switching to a new height three minute measurements are always excluded
(stabilization period). All sampling heights of the tower are equipped with meteorological sensors (wind speed and direction,
air pressure and temperature, relative humidity).

The **Observatoire Pérenne de l'Environnement** (OPE, 48.563° N, 5.506° E, 395 m a.s.l.) is located on the eastern edge of
the Paris Basin in the North Eastern part of France. The station is located in a rural area with large crop fields, and some



pastures and forest patches. A local village and small roads are about one kilometre away. The closest large towns are between 30 km and 40 km away and a major road at about 15 km distance. The station hosts a complete set of in situ measurements of meteorological parameters, trace gases ($CO_2$, $CH_4$, $N_2O$, CO, $O_3$, NOx, and SO2) and particle characteristics. The station is

part of the French aerosol in situ network contributing to ACTRIS and of the IRSN (Institut de Radioprotection et de Sûreté Nucléaire) network for the ambient air radioactivity monitoring. It also contributes to the French air quality monitoring network and to the European Monitoring and Evaluation Program. The infrastructure, including a 120 m tall tower, was built in 2009 and 2010 and the various measurements started between 2011 and 2013. Ambient air is sampled at three levels, 10 m, 50 m and 120 m a.g.l. of the tower and is analysed by Picarro Cavity Ring-Down Spectrometers (CRDS) series G1000 and G2000

for $CO_2$, $CH_4$, $H_2O$ and CO as well as Los Gatos Research Off-Axis-ICOS-spectrometers for $N_2O$ and CO (Conil et al., 2019). Sampling period for each level is 20 minutes, including an automatic rejection of the first five minutes. Meteorological parameters are measured at all air sampling levels.

The ICOS station **Hohenpeißenberg** (HPB, 47.801° N, 11.010° E, 934 m a.s.l.) is located on top of a solitaire hill that rises

approx. 300 m above the almost flat to rolling landscape, 30 km north of the Alps and approx. 60 km southwest of Munich. Main land uses are forests and meadows. The station is hosted by a concrete television tower operated and managed by the DFMG. Cities closest to the station are Weilheim (10 km east of the station, 20 000 inhabitants), Landsberg (30 km north of the station, 30 000 inhabitants), Augsburg (60 km north of the station, 270 000 inhabitants), Munich (60 km northeast of the station, one million inhabitants) and Innsbruck (65 km south of the station, 127 000 inhabitants). Air inlets are at 50 m, 93 m,

and 131 m. A Picarro G2401 CRDS analyser, measuring $CO_2$, $CH_4$, CO, and a Los Gatos Research 913-0015 (EP) OA-ICOS analyser, measuring CO and $N_2O$, are installed in the basement of the tower. Air is being sampled for five min from each level where data for the first minute after switching are discarded. All inlet lines are continuously flushed with approx. 5 L min$^{-1}$. Meteorological sensors (air temperature, relative humidity, wind speed and direction) are installed at every sampling height.

## 2.3 Atmospheric transport modelling for ICOS stations

A footprint simulation tool based on the regional atmospheric transport model STILT (Stochastic Time Inverted Lagrangian Transport; Lin et al, 2003; Gerbig et al., 2006) was implemented at the ICOS Carbon Portal (https://www.icos-cp.eu/about-stilt) as a service for ICOS station PIs and data users. The STILT model simulates atmospheric transport by following a particle ensemble released at the measurement site backward in time and calculates footprints that represent the sensitivity of tracer concentrations at this site to surface fluxes upstream. The footprints are mapped on a 1/12° latitude x 1/8° longitude grid and

coupled to the EDGAR v4.3 emission inventory (Janssens-Meanhout et al., 2019) and the biosphere model VPRM (Mahadevan et al., 2008) to simulate atmospheric $CO_2$ and CO concentrations. These regional concentration components represent the influence from surface fluxes inside the model domain (covering the greater part of Europe). For $CO_2$ the contributions from global fluxes are accounted for by using initial and lateral boundary conditions from the Jena CarboScope global analysed $CO_2$ concentration fields (http://www.bgc-jena.mpg.de/CarboScope/s/s04oc_v4.3.3D.html), while for CO only regional

contributions are evaluated in our study.

## 2.4 The automated ICOS flask sampler

The automated ICOS flask sampler was designed and constructed at the Max Planck Institute for Biogeochemistry (MPI-BGC), Jena, Germany, by the Flask and Calibration Laboratory (FCL) of the CAL to allow automated air sampling under highly standardized conditions. The sampler can hold up to 24 individual glass flasks (four drawers with sic flasks each) for

separate air sampling events (Fig. 2, upper panel). The glass flasks can be individually replaced and sent to the CAL for analysis. The glass flasks used within ICOS (three litre volume, product code ICOS3000 by Pfaudler Normag Systems GmbH, Germany) were developed according to ICOS' specific requirements based on well-proven designs (Sturm et al., 2004). Each



flask has two valves, one at each end, that allow air exchange by flushing sample air through the flask. The flasks are attached with ½" clamp-ring connectors to the flask sampler. The flask valves with PCTFE sealed end-caps can be motor-driven opened
and closed.

A sample is taken by flushing air through a flask at a constant over-pressure of 1.6 bar (absolute).  Sampling at over-pressure increases the amount of available sample air for analysis and allows detecting flasks with leak problems. Flasks are pre-filled with 1.6 bar of dry ambient air with a well-known composition at the CAL to avoid concentration changes due to wall
adsorption effects. The schematic sampler layout is depicted in Fig. 2 (lower panel). Incoming air is dried to a dew-point of approx. -40° C by passing a cooled glass vessel where the exceeding humidity is frozen out. The glass vessel is placed in a silicon oil heat bath that is cooled for drying and heated for out-flushing the collected water to regenerate the trap. The drying unit is automated and consists of two independent inter-switchable drying branches that complement each other and allow a near interruption-free drying. The dryer design is inspired by an already existing system from Neubert et al. (2004). The
incoming sample air is compressed with a pump (Air Dimensions J161-AF-HJ0). A mass-flow controller (MFC, Bronkhorst F-201CV) between compressor and flasks allows to sample at pre-set flow rates, i.e. with a decreasing flow rate over time so that the sample represents a real average e.g. over one hour (Turnbull et al., 2012). The flask pressure during sampling (1.6 bar) is kept constant through a pressure regulator at the outlet of the flasks.  An over-pressure valve set at (2.0 bar) behind the pump assures a constant flow rate through the intake line, independent of the flow rate through the mass flow controller.


In the case of ICOS we strive to sample real one-hour mean concentrations in 3-Litre flasks. The 1/t filling approach requires for this specific case a theoretical dynamic flowrate between 80 mL min$^{-1}$ and infinity. In reality, the maximum flow rate of the selected flow controller is limited to 2 L min$^{-1}$. Therefore, the in situ measurement is averaged with the weighting function resulting from the real flow through the sampled flask. To overcome the flow limitations in the first minutes, the flask is purged
for 30 minutes prior to sampling, assuring a complete air exchange. Average concentrations with the aimed uncertainty can only be reached under sufficiently stable concentration conditions during sampling. For a hypothetical ambient $CO_2$ variability of 1 ppm the upper limit of the associated $CO_2$ flask sample concentration uncertainty was estimated to be in the order of 0.1 ppm.

With the current design of the flask sampler, technical restrictions do not allow parallel sampling of flask duplicates or triplicates as a means for quality control e.g. based on flask pair agreement. The technical effort to allow exact parallel hourly averaged sampling is very high. Therefore, the ICOS Atmosphere Monitoring Station Assembly (MSA) decided to sample only single flasks. This seems appropriate because in the ICOS network the flask sampler is always collecting flasks in parallel to continuous measurements, and erroneously collected flasks or errors due to flask leakages can be detected when comparing
results with the continuous data. Therefore, in contrast to general practice of duplicate flask sampling, in our network single flask sampling seems to be sufficient to meet ICOS objectives. This has the additional advantage that single flask sampling allows more frequent sampling and thus a more representative coverage of the footprint of the stations. If true duplicate samples are required in the future, the flask sampler is designed to accommodate an additional mass flow controller to fulfil this task. The sampler is controlled by an integrated computer offering a broad range of interaction possibilities satisfying the emerging
needs within ICOS. Sampling event time schemes can be pre-programmed and communication with external devices (i.e. data loggers) is possible with analogue or digital signals. Flask to port attributions are completely barcode controlled. Sampling and sensor data are automatically stored and all necessary sampling-related data can be automatically transferred to the CAL. Various automated internet assisted approaches like remote programming of sampling times and pre-selection of samples are possible.



### 3 Aims and technical constraints of ICOS flask sampling

As briefly outlined above, there are three main aims for regular flask sampling at ICOS stations:

1. Flask results are used for comparison with in situ observations (i.e. $CO_2$, $CH_4$, CO, ($N_2O$)). This comparison provides an ongoing quality control of the in situ measurement system, including the intake lines.

2. Flasks are analysed for components not measured continuously at the station, such as $SF_6$ or $H_2$, but also stable isotopes of $CO_2$ or $O_2/N_2$ ratio. The aim is here to monitor large-scale representative concentration levels of these components, which allow estimating their continental fluxes with help of inverse modelling.

3. A subset of flasks are analysed for $^{14}C$ in $CO_2$ to allow determining the atmospheric fossil fuel $CO_2$ component (ff$CO_2$) and with help of these data and inverse modelling to estimate the continental fossil fuel $CO_2$ source strength of the sampled areas.

To meet aims 1 and 2 flask sampling during well-mixed meteorological conditions is required and the sampled footprints should not be dominated by particular hot spot source areas. Particularly for aim 2, we further strive at covering the entire daytime footprint of the station. In contrast, aim 3, due to the generally small fossil fuel signals at ICOS stations, requires targeted sampling of "hot spot emission areas" in the footprint to maximize the fossil fuel $CO_2$ signal in the samples. Note that the detection limit (or measurement uncertainty) of the fossil fuel $CO_2$ (ff$CO_2$) component with $^{14}CO_2$ measurements is of order 1-1.5 ppm (e.g. Levin et al., 2011).

There are a number of technical/logistic constraints concerning flask sampling, shipment and analysis in ICOS, which need to be taken into account when designing an operational sampling strategy that best meets the three aims listed above. The most important limitations are listed in the following:

*Timing:* In order that all flask sample results are useful for flux estimates with current regional inversion models, flasks should be collected during mid-day or early afternoon at the standard ICOS tall tower stations. During this time of the day, atmospheric mixing is strong and model transport errors are smaller than during night (Geels et al., 2007). For all samplings, wind speeds should be larger than about 2 m s$^{-1}$, so that the sampled footprint is well defined. The strategy outlined below has been developed for tall tower sites that are located not directly at the coast, i.e. that are of predominantly continental character.

*Intake height:* There is only one intake line from the highest level of the tower running to the flask sampler; therefore, only the continuous observations from this height can be quality-controlled with parallel sampled flasks (aim 1). As modellers prefer using data (aim 2) from the highest level of the tower (largest footprint, most representative, etc.), all flasks will be sampled from that highest level (as specified in the ICOS Atmospheric Station Specification Document, https://icos-atc.lsce.ipsl.fr/filebrowser/download/69422).

*Integration period:* Flasks should be sampled as integrals, i.e. the collected sample should represent a real mean of ambient air (e.g., a 1-hour mean, comparable to current model resolution). Also, synchronizing in situ continuous observations and integrated flask sampling is important for the quality control aim (aim 1). This latter requirement is easier to achieve with longer integration times in flask sampling. This means, however, that for comparison reasons, the continuous in situ observations must be kept at the flask sampling height during the entire flask sampling period (i.e. no calibration gas measurement, no switching of in situ intake heights during flask sampling, no profile information available). This also means that flow rates, delay volumes and residence times in the tubing, as well as time of both, flask and in-situ sampling systems must be properly monitored.





*Flask handling:* Flasks need to be installed and removed manually from the sampler. Remote stations are regularly visited about once per month by a technician. The flasks sampled to meet aim 1 should be shipped to the FCL within one month after sampling, so that a potential bias between in situ and flask analyses is detected without major delay. $^{14}CO_2$ analysis of flasks

in the CRL is less urgent, therefore a few months delay in shipment of flasks collected for aim 3 are acceptable.

*CAL measurement capacity:* While the capacity for flask analysis at the FCL has been designed for a total of about 100 flask analyses per station per year, the capacity for $^{14}CO_2$ analyses in the Central Radiocarbon Laboratory (CRL), which are performed *after* analysis of all other components at the FCL, is only about ¼, i.e., on average, 25 samples per station per year.

Consequently, all flasks will be shipped from the station to the FCL and after analysis a subset will be shipped for further analysis to the CRL. After all analyses have been finished all flasks including those, which were analysed at the CRL are leak-tested and conditioned at the FCL before dispatch to the stations.

**4 Results**

**4.1 Solutions and testing to meet aim 1: Ongoing quality control**

The ICOS atmospheric station network supported by ICOS Central Facilities (ATC and CAL) has been designed and implemented to achieve the highest possible accuracy, precision and compatibility of atmospheric GHGs measurements. ICOS aims to meet the compatibility goals agreed on by the international WMO/GAW measurements community (WMO, 2018) for all its measured components. These compatibility goals were chosen by the community to detect small inter-station gradients and to be used to estimate flux distributions by means of inverse models. For ICOS $CO_2$ observations, a compatibility goal of

0.1 ppm or better is compulsory. Similarly, ICOS needs to meet the WMO compatibility goals for $CH_4$ and CO, which are 2 ppb for both gases (WMO 2018). First evaluations of ICOS $CO_2$ measurements indeed yield monthly mean afternoon differences between stations in the free troposphere above 100 m of typically very few ppm (Ramonet et al., 2020), underlining the importance of excellent precision and compatibility of these observations.

With a regular and frequent comparison of flask and in situ measurements, ICOS aims at independently monitoring their compatibility and provide respective alerts if e.g. the average difference of $CO_2$ exceeds 0.1 ppm over a few weeks comparisons. Using flasks sampled from a dedicated intake line to cross-check the in situ measurements is an important part of the ICOS quality management. It allows an independent end-to-end QC of the entire in situ measurement system consisting of inlet system, drier, analyser and calibration. As mentioned above, for logistical reasons, about once per month or every five

weeks a box with 12 flasks is scheduled to be shipped from a remote station to the FCL. After analysis, the flask results covering about one month of time will be compared with the corresponding in situ data. In the following paragraph we elaborate the minimum number of comparison flasks and the corresponding time delay to detect a significant $CO_2$ bias between flask and in situ measurements larger than 0.1 ppm. Therefore we tested experimentally at the ICOS pilot station in Heidelberg the envisaged flask sampling procedure, and present here its first application at an ICOS field station.

**4.1.1 Flask – in situ $CO_2$ comparisons in Heidelberg**

Similar to the official ICOS atmosphere stations, Heidelberg is equipped with an ICOS-conform CRDS instrument continuously measuring $CO_2$, $CH_4$ and CO in ambient air. Also the Heidelberg instrument is calibrated with standard gases provided by the FCL and its continuous data are automatically evaluated at the ATC. All flasks have been analysed at the FCL. However, since the site does not have a high tower and is located in an urbanized environment, the variability of the signal

can complicate the flask-in situ comparison.



In order to collect a real hourly integrated air sample in the flask, the flow rate through the flask has to be adjusted during the filling process (Turnbull et al., 2012, see Sect. 2.4). First tests with a decreasing (1/t) flow rate through the flasks were conducted in Heidelberg during the period of September 2018 to February 2019, and with a better suited flow controller for

the 1/t decreasing flow rate from May to October 2019. Ambient air for continuous measurements as well as for flask sampling was collected via by-pass from a permanently flushed intake line from the roof of the institute's building about 30m above local ground. These flasks have been collected not only at low ambient air variability during afternoon hours, but also during other times of the day, when within-hour concentration variations for $CO_2$ at this urban site were higher than 10 ppm. The results of the concentration differences between in situ and flask measurements for $CO_2$ are displayed in Fig. 3 (left panel).

During the first experimental period we obtained three outliers, where flask $CO_2$ results have been up to more than 3 ppm higher than the in situ measurements. $CH_4$ and CO in the flasks (not shown) did, however, compare very well within a few ppb with the continuous in situ data. Although one of the mass flow controllers had some problems to exactly regulate the flow over the large range of flow rates, we did not find obvious reasons for malfunction of the sampling system. The only explanation for the outliers may, thus, be contamination of these flasks with room air, which is elevated in $CO_2$ but not in $CH_4$

or CO compared to outside air.

If we disregard the three outliers in the first testing period (one at a low variability situation, see Fig. 3, right panel) and consider only observations with ambient air $CO_2$ variability < 0.5 ppm, the limited results from the (polluted) Heidelberg site give confidence that flask samples collected over one hour at low ambient $CO_2$ variability are well suited to meet our aim 1 of

ongoing quality control at Class 1 stations. It is important though that the different air residence times in the intake systems of flask sampler and in situ instrument are properly adjusted; they may significantly differ, e.g. if a mixing volume system is installed in the intake lines (as at Hyltemossa). The mean differences between in situ and flask measurements for $CO_2$ in Heidelberg have been 0.02 ppm at an ambient $CO_2$ variability of less than 0.5 ppm, with a standard deviation of ±0.06 ppm (n=18) (see also Fig. 3 right panel, which shows that only one out of the 18 low-variability comparisons lies outside the ±0.1

ppm compatibility range indicated by the dashed red lines). For $CH_4$ we observed for ambient variability smaller than 10 ppb a mean difference of 0.18 ppb with a standard deviation of 0.74 ppb (n = 111). CO comparison data have not been evaluated here as the CRDS in situ data were not finally calibrated and thus not fully compatible with the flask results.

The test measurements in Heidelberg did clearly show that meaningful quality control results can best be obtained during

situations of low ambient concentration variability. Individual concentration differences increase with increasing ambient variability within the one-hour comparison period. The reason for this increase may be uncertainties in the synchronization of the measurements (note that a few minutes shifts in the timing of the integration already introduces a significant bias) or also due to incorrect flow rates through the flasks in the 1/t sampling scheme. For the QC aim, flask samples should preferentially be collected during low variability situations. We therefore evaluated how frequent afternoon events with less than 0.5 ppm

variability occur at typical ICOS stations. In the years 2016 to 2019, except for few stations and for few summer months, we find at all five stations at least 10 hours per month at mid-day (13 h local time (LT)) with hourly $CO_2$ standard deviations smaller than 0.5 ppm. On average over the year more than half of all midday hours had $CO_2$ standard deviations below 0.5 ppm. Based on this evaluation, we decided that we will not need to pre-select sampling days with low ambient variability but can pursue a very simple sampling scheme, e.g. sampling every three or four days, to be able to detect a mean bias larger than

0.1 ppm between flask and continuous measurements within a period of 4-5 weeks. On average we can expect that every second flask we sample is suitable for precise intercomparison with in situ measurements. This simple methodology will help us meeting aim 2 (see below).





**4.1.2 Flask – in situ $CO_2$ comparisons at the ICOS station Hohenpeißenberg**

A very first field test of our flask sampling scheme for QC was conducted at the ICOS station Hohenpeißenberg (HPB). From the highest level of the tall tower (131 m) ambient air for continuous measurements as well as for flask sampling was collected via two separate lines. Collecting flasks at HPB started in July 2019. The flasks were always sampled with a decreasing 1/t flow rate and between 12:30 and 14:00 UTC as we aim for conditions with low ambient variability, which is largest in well

mixed conditions during the afternoon. Up to now, 48 flasks have been collected, which could be used for QC of this ICOS Class 1 station. The overall results of the concentration differences for $CO_2$ for the complete test period are shown in Fig. 4 (left panel).

Our first results of the comparison between continuous measurements and flasks were available in October 2019 and showed

larger differences between in situ and flasks measurements than expected. A mean difference of 0.33 ppm with a standard deviation of ±0.13 ppm (n=4) was determined for situations with an ambient variability of less than 0.5 ppm. Based on these results the intake system and the entire $CO_2$ instrumentation was carefully checked. Whilst the last regular leak test on April 10, 2019 passed the ICOS specifications, an unscheduled leak test was performed at the end of October 2019, following up the unexpected flasks results. During this test, a leak in the 131 m sampling line to the instruments for the continuous

measurements was detected in the shelter. The leak was eliminated on October 30, 2019, and leak tightness was confirmed by a second leak test on November 19, 2019.

For the period after the leak elimination, the calculated differences between in situ and flask measurements for an ambient variability of less than 0.5 ppm lie all within the compatibility goal for $CO_2$ (0.1 ppm), see blue dots in Fig. 4 (right panel).

The mean difference between flasks and in situ measurements is 0.01 ppm with a standard deviation of ±0.06 ppm (n=5). These results of the first field test of the flask sampling scheme for QC are promising, e.g. enabling detection of potential leaks at the stations. Once the flask QC procedures have been set up operational, potential system malfunctions can be detected within a month, complementing the half-yearly compulsory ICOS leak tests.

**4.2 Solutions and testing to meet aim 2: Representative flask sampling**

In the preceding section we could show that low ambient variability situations would be best suited to meet aim 1. Moreover, a potential bias between flask and in situ measurements could be detected with better confidence with an increased number of comparisons. However, for meeting aim 2, a scheme collecting flasks only during low variability situations may cause a significant bias in the sampled footprint. We have tested if such a sampling bias would be visible in the European ICOS network and calculated with STILT all afternoon (13 h LT) footprints of the five selected stations for the year 2017. Figure 5

shows respective aggregated footprints for the month of October 2017. The left panels in each of the five rows show the aggregations if every afternoon hour (13 h LT) was sampled, the middle panels the aggregated footprints for every third day and the right panels show the 10 footprints with the lowest variability during this month. As expected, we can see that regional coverage of the entire station footprint is generally better when sampling randomly every third day than when sampling the 10 days with the lowest variability.


In addition to the footprint analysis, which gives a visual qualitative idea of the effect of different flask sampling schemes, we evaluated the first three years of continuous $CO_2$ measurements from the five ICOS stations to quantify the effect of random sampling every three days versus sampling only low variability situations. Figure 6 shows, in the upper panels for each station, all available hourly atmospheric $CO_2$ data as grey dots, while the blue lines, each shifted by one day, connect the 13 h LT data

every three days. The red dots in the upper panels highlight the 10 lowest variability afternoon values in each month. As expected, all summer afternoon concentrations generally fall into the lower concentration range of the bulk of data. At all





stations, the variability changes from a diurnal shape during the summer months to a more synoptic variability in the winter half-year (for more details see also Fig. 8 and 9). This synoptic variability is also represented in the afternoon sampling. In the five middle panels of Figure 6 we have plotted as black dots monthly means calculated from all afternoon hours between 11h and 15 h LT as well as their standard deviations. The blue dots show the monthly mean values obtained from sampling every third day (the three different 3-day patterns are shown in individual shifted blue lines), while the red dots represent the monthly means calculated from the 10 samples with the lowest variability (the coloured dots were shifted by one day each for better visibility). It is obvious that regular sampling provides much more representative monthly means, deviating only in few cases from the all-afternoon means in $CO_2$ by more than 2 ppm (Fig. 6, lowest panels). If samples were collected at low variability only, they would often underestimate monthly mean values, in some cases by more than four ppm (red lines in Fig. 6 lowest panels). Although also regular sampling every third day introduces some variable deviations from the correct afternoon means, sampling only at low variability may introduce rather large biases mainly towards lower $CO_2$ concentrations. Note that inversion models also select measured data for their inversion runs only for time of the day and not for low variability data to estimate fluxes (Rödenbeck, 2005).

We have investigated here only potential sampling effects on $CO_2$ concentrations, however, also other tracer concentrations are expected to be affected in a similar way. For the ICOS atmosphere network we, therefore, choose the simpler sampling scheme of one flask every third day. This sampling scheme is expected to serve aims 1 and 2, where those flasks with low within-hour variability (on average one flask per week, see Sect. 4.1) could be used for the quality control aim, while all flask samples would deliver as much as possible representative data for all additional trace components analysed in the FCL solely on flasks.

**4.3 Solutions and testing to meet aim 3: Catching potentially high fossil fuel $CO_2$ events**

First [14]C analyses on integrated $CO_2$ samples at ICOS stations showed rather low average fossil fuel $CO_2$ (ff$CO_2$) concentrations, therewith confirming that ICOS stations primarily monitor the terrestrial biospheric signals. Figure 7 (upper panels of the graphs for the individual stations) shows our first [14]$CO_2$ results from the two-week integrated $CO_2$ sampling at Hyltemossa, Křešín, Observatoire Pérenne de l'Environnement and Hohenpeißenberg. Particularly during summer, the monthly mean regional fossil fuel $CO_2$ offsets, if compared to a background level calculated from the composite of two-week integrated [14]$CO_2$ measurements at Jungfraujoch in the Swiss Alps and Mace Head at the Irish coast, are often lower than a few ppm (Fig. 7, lower panels). Only during winter, regional ff$CO_2$ offsets can reach two-week mean concentrations of more than 5 ppm. These signals, although providing good mean ff$CO_2$ results for the average footprints of the stations, are often too small to provide a solid top-down constraint of regional fossil fuel $CO_2$ emission inventories and its changes when evaluated in regional model inversions (Levin and Rödenbeck, 2008; Wang et al., 2018). One of the aims of flask sampling in ICOS is, therefore, to explicitly sample air, which has passed over fossil fuel $CO_2$ emission areas. Ideally we would like to obtain signals and analyse flasks for [14]$CO_2$ only in cases when the expected fossil fuel $CO_2$ component is larger than 4-5 ppm. This would allow to obtain an uncertainty of the estimated ff$CO_2$ component below 30 % (Levin et al., 2003; Turnbull et al., 2006). Further, as sample preparation for [14]C analysis is very laborious and the capacity of the CRL is limited to about 25 flask samples per station per year, one should know beforehand, if a sample potentially contains a significant regional fossil fuel $CO_2$ component. This could either be found out with Near Real Time transport model simulations or directly using the in situ observations at the station.

A good indicator for the potential regional fossil fuel $CO_2$ concentration at a station is the ambient CO concentration (Levin and Karstens, 2007), a trace gas that is monitored continuously at all ICOS Class 1 sites. It would then depend on the average CO/ff$CO_2$ ratio of fossil fuel emissions in the footprint of the stations to estimate from measured CO the expected ff$CO_2$





concentration. Mean $CO/ffCO_2$ emission ratios can be very different in different countries, they mainly depend on the energy

production processes and on domestic heating systems. In this respect, also the share of biofuel use may be relevant. In our

study we first analysed our selected ICOS stations for regional fossil fuel $CO_2$ signals larger than 4 ppm, and determined the

frequencies of those events. Note that, in order for the flask results to be used in transport model investigations, similar to all

other flask samples, also $^{14}CO_2$ flasks should be collected during early afternoon when atmospheric mixing can be modelled

with good confidence. During these situations, however, any $ffCO_2$ signals will be highly diluted. Similar to the approach in

the previous section, we investigated the potential $ffCO_2$ levels for the five stations Hyltemossa, Gartow, Křešín, Observatoire

Pérenne de l'Environnement and Hohenpeißenberg, first theoretically with STILT model simulations transporting

EDGARv4.3 emissions to the five measurement sites. As a second step, we evaluated real continuous $CO_2$ and CO observations

from 2017 and 2018 (see Table 1).

### 4.3.1 Investigation of afternoon fossil fuel $CO_2$ events in 2017 at Gartow

Figure 8 shows in the upper two panels ambient STILT-simulated $CO_2$ and CO mole fractions at Gartow 341 m in July 2017

(13 h LT values highlighted by coloured symbols), while the third panel compares STILT-simulated total $CO_2$ (blue line) to

observations (black line). The agreement between model and observations turned out to be reasonable, particularly during

afternoon hours. In this summer month, deviations of the model simulations from observations are larger during night, when

the model seems to underestimate the measured concentration pile-up. This model deficiency is the reason why we decided to

collect the flask samples at midday or in the afternoon, making sure the data can be used in inversion estimates of fluxes. In

the fourth panel of Fig. 8 the simulated regional $CO_2$ components ($ffCO_2$ offset and biospheric $CO_2$ offset) originating from

fluxes in the model domain (covering the greater part of Europe) are displayed, underlining the generally moderate fossil fuel

$CO_2$ signal at Gartow in July. Indeed, summer situations with potentially high $ffCO_2$ concentrations are rare (1-5 cases) at all

ICOS stations, and at Gartow only during three days, i.e. on July 1, 7 and 27, modelled afternoon $ffCO_2$ was larger than 4 ppm

(highlighted by red crosses in the upper panel of Fig. 8). At the same time, the modelled CO offset was elevated, but did not

reach 0.04 ppm (second panel). CO offsets were estimated relative to the minimum modelled CO concentration of the last

three days (grey line in second panel). In October 2017, the modelled (Fig 9, upper panel) and measured CO (Fig. 9, lowest

panel) offsets do, however, rather frequently exceed 0.04 ppm. The generally good correlation between simulated $ffCO_2$ and

CO offset can therefore be used as a criterion for $ffCO_2$ in collected flasks, and 0.04 ppm may be a good threshold for Gartow

to predict a $ffCO_2$ signal of more than 4 ppm in sampled ambient air. This is supported by real observations displayed in the

two lowermost panels of Figs. 8 and 9, where observed CO offsets > 0.04 ppm (marked by magenta crosses) coincide with

high total $CO_2$, and also with STILT-simulated $ffCO_2$ (see for example the synoptic event on October 18-19, 2017).

The aggregated footprints of the three afternoon situations with STILT-simulated $ffCO_2$ > 4 ppm in July 2017 are displayed

in Fig. 10 (upper panels). They show south-westerly trajectories and a dominating surface influence from the highly populated

German Ruhr area, but also some influences from large emitters (e.g., power plants) in north western Germany and at the

Netherland's North Sea coast. The main influence area with high $ffCO_2$ emissions in October 2017 (Fig. 10, lower panels),

show also Berlin as a significant emitter and some "hot spots" close to the German-Polish border in the south-east.

### 4.3.2 Investigation of afternoon fossil fuel $CO_2$ events in 2017 and 2018 at Hyltemossa, Křešín, Observatoire Pérenne
### de l'Environnement and Hohenpeißenberg

Overlapping measurements and STILT model runs are also available for the other four ICOS stations. The general picture is

similar here as in Gartow, but the number of elevated $ffCO_2$ events is often even smaller at these stations than at Gartow. For

example we find no $ffCO_2$ events at HTM, GAT, KRE and HPB, and only three at OPE in July 2018 (Table 1). Simultaneously

observed CO elevations relative to background are often only small in summer and do not reach the (preliminary) threshold of





0.04 ppm. Starting in October or November, ffCO$_2$ elevations become more frequent coupled to the more synoptic variability of GHGs in the winter half-year (cf. Fig. 6 upper panels). The number of modelled fossil fuel CO$_2$ events larger than 4 ppm for all months in 2017 and 2018 or based on observed CO offsets larger than 0.04 ppm using the same estimate for the CO background as for the model results displayed in Fig. 8 and 9 are listed in Table 1. Only in the winter half year we can potentially sample well measurable fossil fuel CO$_2$ signals. Lower $\Delta$CO thresholds could be used for summer, then accepting

larger uncertainties of the ffCO$_2$ component. A better alternative would probably be to restrict $^{14}$C analysis on flasks collected in autumn, winter and spring, with the additional advantage that the variability of biospheric signals is smaller during these seasons (cf. Fig. 9).

To give some indications of the main ffCO$_2$ emission areas influencing the four stations, Fig. 11 shows aggregated footprints

as well as the respective surface influence areas contributing to modelled ffCO$_2$ concentrations larger than 4 ppm in October 2017. At all four stations and also at Gartow (Fig. 10) the areas potentially contributing significantly to the fossil fuel signals are located rather far away and many of them are associated to large coal-fired power plants or other point sources. But also a few big cities such as Prague at Křešín occasionally contribute.

**5 Implementation of the flask sampling scheme at ICOS stations**

Sampling one flask every third day, independent of ambient CO$_2$ variability can easily be implemented at ICOS stations since sampling of all 24 flasks in the sampler can individually be programmed in advance. Assuming that flasks can be exchanged about once per month, during this time span 12 flasks would have been collected and could then be shipped in one box to the FCL for analysis. The remaining 12 flasks in the sampler would be reserved for ffCO$_2$ event sampling. In order to have a realistic chance to catch all possible events at a station, the sampler would be set to fill one of these flasks on each day between

the regular every third day sampling. As continuous trace gas measurement data are transferred from the station to the ATC every night, level-1 CO data are available on the next morning after flask sampling the day before. These data will then be automatically evaluated at the ATC for potentially elevated CO to decide if the flask that had been collected on the day before has potentially an elevated ffCO$_2$ concentration and should be retained for $^{14}$CO$_2$ analysis. If yes, the flask sampler will obtain a respective message from the ATC. If not, the flask can be re-sampled. Based on our analysis of modelled ffCO$_2$ for the year

2017 and 2018, the likelihood is small that more than 12 ffCO$_2$ events are sampled within one month. Also, some of the events may already have been sampled in one of the "regular" every third day flasks. If this has been the case, these flasks will be marked, so that they are passed on to the CRL after analysis of all other components in the FCL. In the future, especially the flask sampling strategy for ffCO$_2$ events might change once real-time GHG prediction systems or prognostic footprint products are available, which would allow more accurate targeting of certain emission areas. First tests, using prognostic trajectories to

automatically trigger $^{14}$CO$_2$ flask sampling are made at the ICOS CRL pilot station and at selected ICOS Class 1 stations, but are not yet mature enough to be implemented in the entire ICOS network. It is, however, also worth to mention that sampling flasks also during night time could largely increase the significance of $^{14}$C-based ffCO$_2$ estimates. Currently we optimize our sampling strategy to meet the inability of transport models digesting also night time data. This situation is unfortunate and must urgently be improved in order to increase our ability to monitor, in a top-down way, long-term changes of the envisaged

ffCO$_2$ emissions in Europe.

**6 Conclusions**

Developing a flask sampling strategy for a network like ICOS is a new approach, which, to our knowledge, has not yet been taken in any other sampling network. It may contribute to optimizing efforts at the (remote) ICOS stations as well as the analytical capacities and capabilities of the ICOS Central Analytical Laboratories. Our strategy was designed to meet, on one



hand, the requirements for quality control, making sure by comparison of flask results with the parallel in situ measurements that ICOS data are of highest precision and accuracy. Our first results showed that this strategy of independent quality control is successfully working. At the same time, our sampling scheme will provide flask results that can be optimally used in current inverse modelling tasks to estimate continental fluxes, not only of core ICOS components, such as $CO_2$ and $CH_4$, but also of trace substances, which are not yet measured continuously. Trying to monitor also fossil fuel $CO_2$ emission hot spots at ICOS

stations during well-mixed afternoon hours will be a particular challenge, because the ff$CO_2$ influence at that time of the day is often very small. There is thus an urgent need for transport model improvement so that also night time data can be used for the inversion of fluxes. Experience of the coming years will show if our current strategy is successful to meet all aims or needs further adaption.

**Code availability**: The Jupyter notebook to perform the analysis of STILT model results and ICOS in situ measurements will be made available at ICOS Carbon Portal upon request.

**Data availability:** ICOS RI: Atmospheric Greenhouse Gas Mole Fractions of $CO_2$, $CH_4$, CO, $^{14}CO_2$ and Meteorological Observations September 2015 - April 2019 for 19 stations (49 vertical levels), final quality controlled Level 2 data (Version

1.0). ICOS ERIC - Carbon Portal, https://doi.org/10.18160/CE2R-CC91, 2019.

**Author contributions:** IL and UK designed the study, UK developed the Jupyter notebook and conducted the STILT model runs, ME built the flask sampler and developed its software, FM and SA conducted the flask sampling and evaluated the comparison data, DR was responsible for flask, SH for $^{14}CO_2$ analysis, MR was responsible for ICOS data evaluation, GV,

SC, MH, DK and ML were responsible for the measurements at the ICOS stations. IL and UK prepared the manuscript with contributions from all other co-authors.

**Competing interests**: The authors declare that they have no conflict of interest.

**Acknowledgements**
This work has been funded by the European Commission in the framework of the RINGO project REP-730944-2. All measurements and model estimates were conducted within the ICOS RI consortium by scientists contributing to the different components (National Networks, Central Facilities and Carbon Portal) that are jointly funded by national funding agencies from all ICOS partner countries. Operation of the Krešín u Pacova station was supported by the Ministry of Education, Youth

and Sports of CR within the CzeCOS program, grant number LM2015061.

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





**Table 1: Number of mid-day (13 h LT) ffCO₂ events > 4 ppm estimated by the STILT model for 2017 and 2018 as well as potential fossil fuel CO₂ events in both years, based on modelled (m) and observed (o) ΔCO elevations of more than 0.04 ppm compared to background (entries are empty if less than 20 afternoon CO observations are available in the respective month)**

| | HTM | | | GAT | | | KRE | | | OPE | | | HPB | | |
|---|---|---|---|---|---|---|---|---|---|---|---|---|---|---|---|
| | ffCO₂ | ΔCO | | ffCO₂ | ΔCO | | ffCO₂ | ΔCO | | ffCO₂ | ΔCO | | ffCO₂ | ΔCO | |
| **2017** | m | m | o | m | m | O | m | m | o | m | m | o | m | m | o |
| Jan | 9 | 2 | | 15 | 7 | | 6 | 8 | | 11 | 10 | 10 | 10 | 15 | |
| Feb | 7 | 4 | | 10 | 6 | | 7 | 7 | | 6 | 3 | 10 | 4 | 9 | |
| Mar | 2 | 0 | | 5 | 4 | | 4 | 3 | | 5 | 5 | 4 | 3 | 3 | 6 |
| Apr | 1 | 0 | | 0 | 0 | | 2 | 0 | | 1 | 0 | 0 | 6 | 0 | 3 |
| May | 0 | 0 | | 1 | 0 | 3 | 4 | 1 | 2 | 2 | 1 | 1 | 2 | 2 | 1 |
| Jun | 0 | 0 | | 2 | 0 | 1 | 0 | 0 | | 0 | 0 | 2 | 1 | 0 | 0 |
| Jul | 0 | 0 | | 3 | 0 | 2 | 0 | 0 | | 0 | 0 | 1 | 0 | 0 | 0 |
| Aug | 0 | 0 | 2 | 2 | 1 | 2 | 0 | 0 | | 0 | 0 | 0 | 0 | 0 | 3 |
| Sept | 2 | 0 | 0 | 6 | 2 | 10 | 3 | 0 | | 3 | 1 | 2 | 1 | 0 | 0 |
| Oct | 1 | 0 | 2 | 8 | 4 | 7 | 4 | 3 | | 1 | 1 | 0 | 1 | 0 | 2 |
| Nov | 5 | 4 | 5 | 13 | 10 | 11 | 5 | 2 | | 5 | 5 | 5 | 9 | 8 | 14 |
| Dec | 3 | 1 | 5 | 8 | 3 | 2 | 10 | 2 | | 2 | 1 | | 9 | 8 | 10 |
| **2018** | | | | | | | | | | | | | | | |
| Jan | 4 | 3 | 7 | 8 | 3 | 8 | 5 | 5 | | 4 | 4 | 8 | 10 | 11 | 6 |
| Feb | 5 | 1 | 7 | 7 | 4 | 11 | 4 | 2 | 16 | 10 | 7 | 13 | 9 | 6 | 17 |
| Mar | 3 | 4 | 8 | 6 | 4 | 11 | 1 | 3 | 16 | 4 | 4 | 7 | 4 | 5 | 9 |
| Apr | 2 | 2 | 3 | 4 | 1 | 2 | 1 | 0 | 3 | 2 | 2 | 1 | 2 | 1 | 0 |
| May | 0 | 0 | 0 | 0 | 0 | 1 | 3 | 0 | 0 | 2 | 2 | 3 | 3 | 2 | 1 |
| Jun | 0 | 0 | 0 | 4 | 0 | 1 | 2 | 0 | 0 | 4 | 2 | 1 | 3 | 0 | 0 |
| Jul | 0 | 0 | 0 | 0 | 0 | 2 | 0 | 0 | 0 | 3 | 2 | 0 | 0 | 0 | 0 |
| Aug | 1 | 0 | 1 | 2 | 0 | 5 | 0 | 0 | 3 | 1 | 1 | 6 | 1 | 0 | 3 |
| Sept | 0 | 0 | 0 | 4 | 1 | 1 | 4 | 1 | 3 | 6 | 3 | 5 | 5 | 2 | 2 |
| Oct | 7 | 2 | 6 | 8 | 3 | 6 | 6 | 1 | 10 | 12 | 7 | 8 | 5 | 3 | 8 |
| Nov | 9 | 5 | 12 | 12 | 7 | | 3 | 4 | 18 | 9 | 6 | 12 | 15 | 15 | 12 |
| Dec | 5 | 4 | 8 | 8 | 2 | | 9 | 6 | 12 | 7 | 7 | 8 | 5 | 9 | 5 |




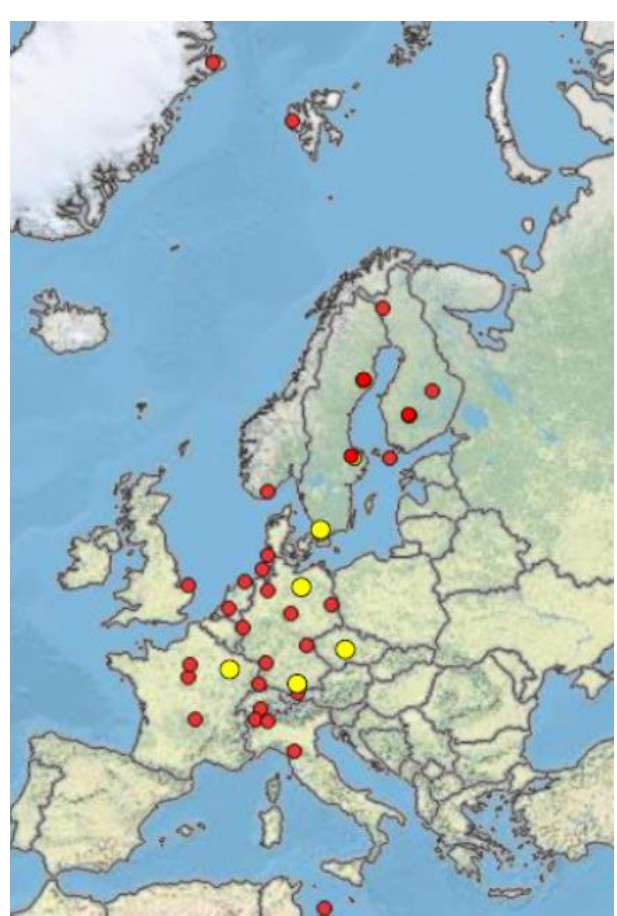

**Figure 1: Map of ICOS atmospheric stations. The five stations included in this study are marked by big yellow dots (© ICOS RI CC**
**BY)**





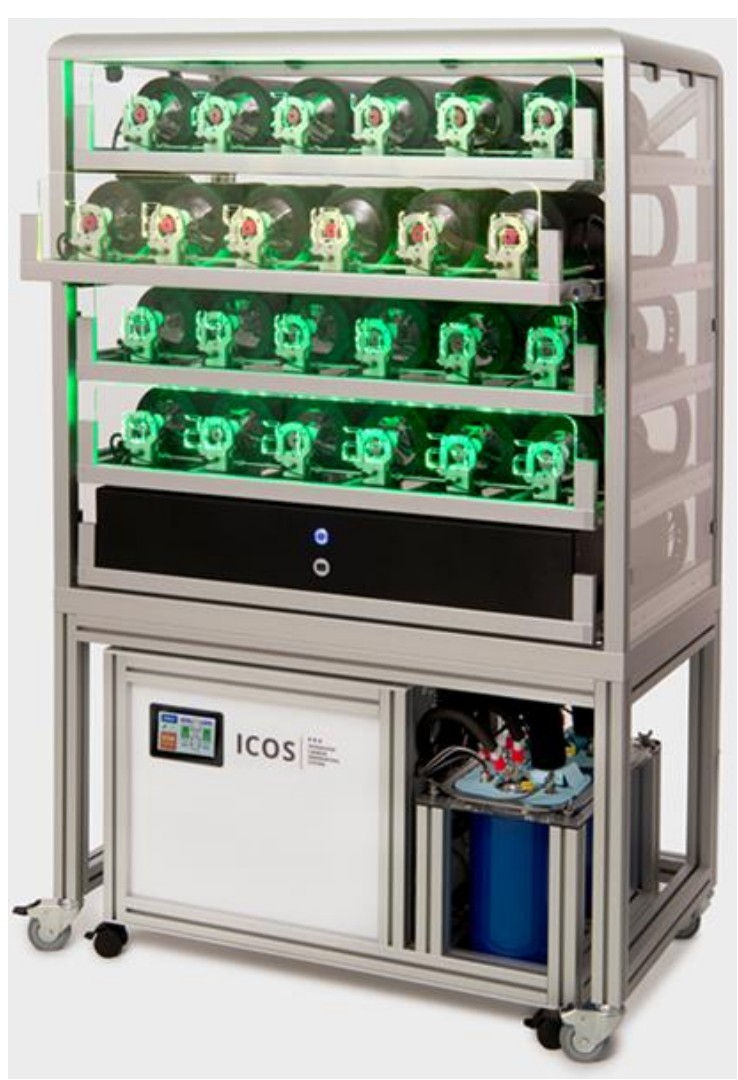

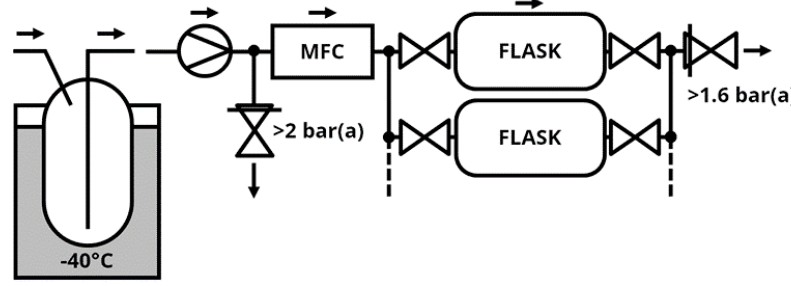


**Figure 2: Photograph of the ICOS flask sampler with schematic flow diagram**






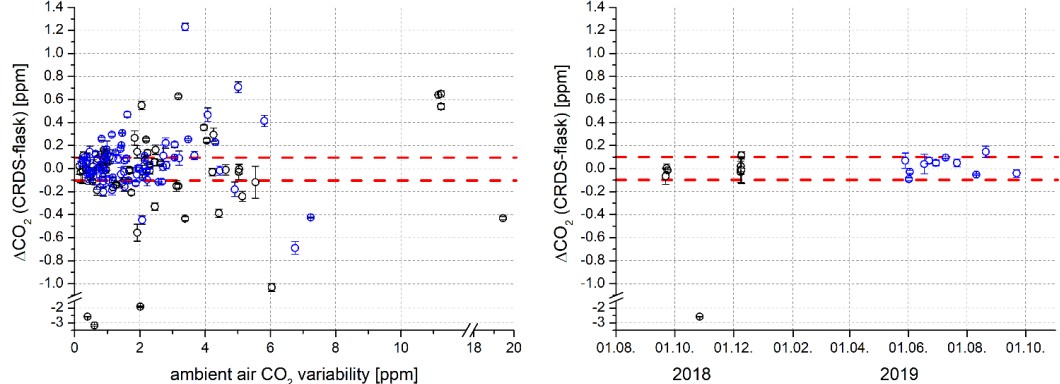


**Figure 3: Left: In situ minus flask CO₂ results obtained with the 1/t flask flushing method to get a real hourly mean sample. Flasks from the second comparison period are marked in blue. Right: Temporal development of the in situ minus flask measurement for ambient air CO₂ variability < 0.5 ppm. All except one of these latter differences lie within the required ±0.1 ppm compatibility range. No sampling was performed between February and May 2019.**


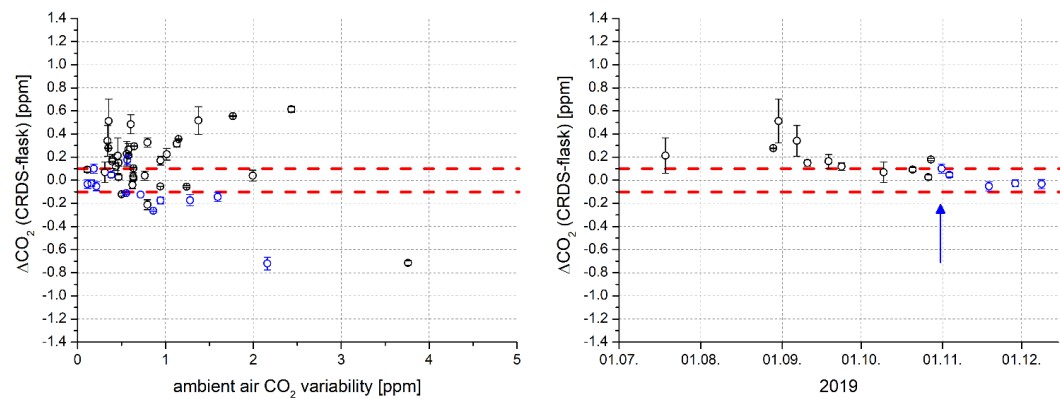


**Figure 4: Same as Fig. 3, but for Hohenpeißenberg. Left: In situ minus flask CO₂ results obtained with the 1/t flask flushing method to sample a real hourly mean sample. Flasks from the comparison period after October 30, 2019 (the date when the leak was sealed) are plotted in blue. Right: Temporal development of the in situ minus flask measurement for ambient variability < 0.5 ppm. All five**
**differences after sealing the leak on October 30, 2019 (blue arrow, blue dots) lie within the required ±0.1 ppm compatibility range.**



**Figure 5: Aggregated footprints calculated for the five ICOS stations from top to bottom: Hyltemossa, Gartow, Křešín, Observatoire Pérenne de l'Environnement and Hohenpeißenberg for October 2017. The left panels show the footprints for all 31 days at 13 h LT, the middle panels the same footprints, but sampled only every third day and the right panels show those of the 10 days with the lowest variability. Note the logarithmic colour scale.**



**Figure 6: CO$_2$ concentration data measured at Hyltemossa (upper left), Gartow (upper right), Křešín (middle left), Observatoire Pérenne de l'Environnement (middle right), and Hohenpeißenberg (lower left). For each station the upper panel shows all hourly data as grey dots while afternoon data (13 h LT) every third day are displayed as three blue lines shifted by one day each. Red dots highlight for each month the 10 afternoon values with the lowest variability. The middle panels show as black dots monthly means and standard deviations of all afternoon hours (11–15 h LT) as well as respective means from afternoon data collected every third day in blue and in red the means of the 10 afternoon values with the lowest variability (for better visibility the coloured dots were shifted by one day each). The low panels present the differences of the selected afternoon means from the respective mean calculated from all afternoon data.**






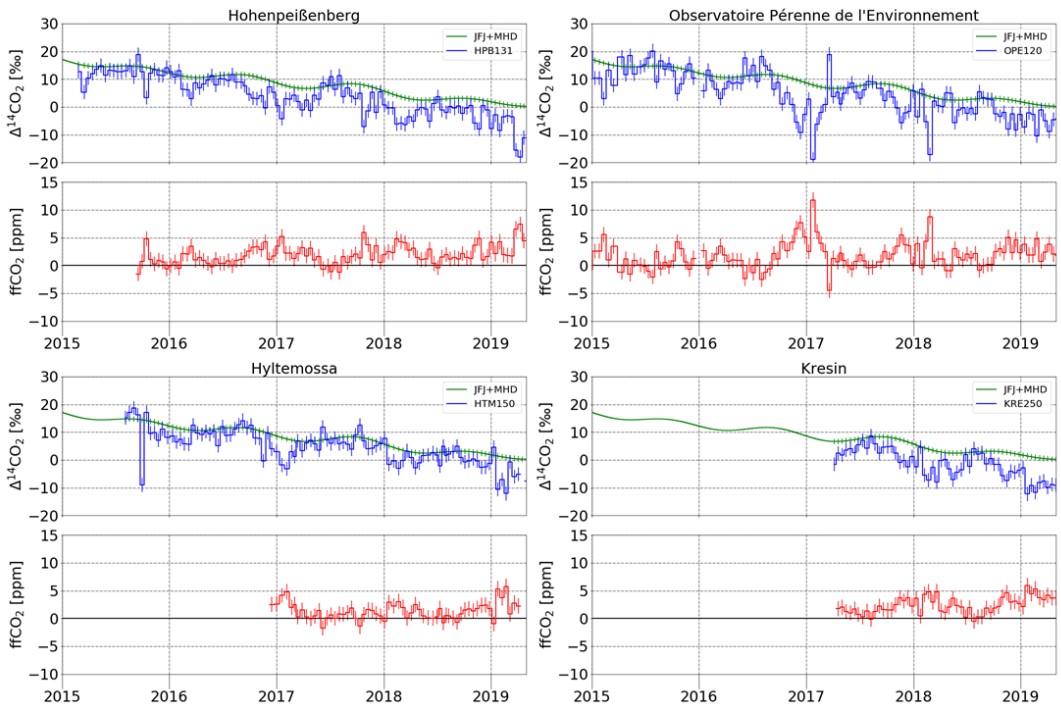

**Figure 7:** **$^{14}CO_2$ observations and estimated fossil fuel $CO_2$ concentrations relative to European background at the ICOS stations**
**Hyltemossa, Křešín, Observatoire Pérenne de l'Environnement and Hohenpeißenberg. The upper panels for each station show the**
**$\Delta^{14}CO_2$ results in permil deviation from the NBS Oxalic acid standard (Stuiver and Polach, 1977) for the respective station (blue**
**histogram) together with the European background, which is estimated as the fit curve to measured data from Jungfraujoch and**
**Mace Head. The lower panels give the regional fossil fuel $CO_2$ offset calculated from the $^{14}CO_2$ and $CO_2$ data according to Levin et**
**al. (2011). For HPB and HTM the ff$CO_2$ calculation starts later than the $^{14}CO_2$ data since for these stations no ICOS $CO_2$ data is**
**available in the early times.**



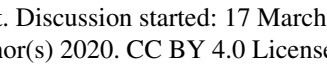

**Figure 8: Variability of STILT-simulated (upper four panels) and measured (lower two panels) CO₂ and CO concentrations at Gartow at the 344/341 m level in July 2017. Afternoon values are highlighted with coloured symbols (blue dots), situations with elevated ffCO₂ based on modelled or measured CO (CO offset > 0.04 ppm) are marked with a magenta cross in the CO and also in the CO₂ records (better visible in Fig. 9 for Oct. 2017 when more such situations occur). CO offsets in STILT model simulations (second panel) and observations (lowest panel) were estimated relative to the minimum CO concentration of the last three days (grey lines).**



Gartow 341m 53.07°N 11.44°E

**Figure 9: Same as Fig. 8, but for October 2017.**




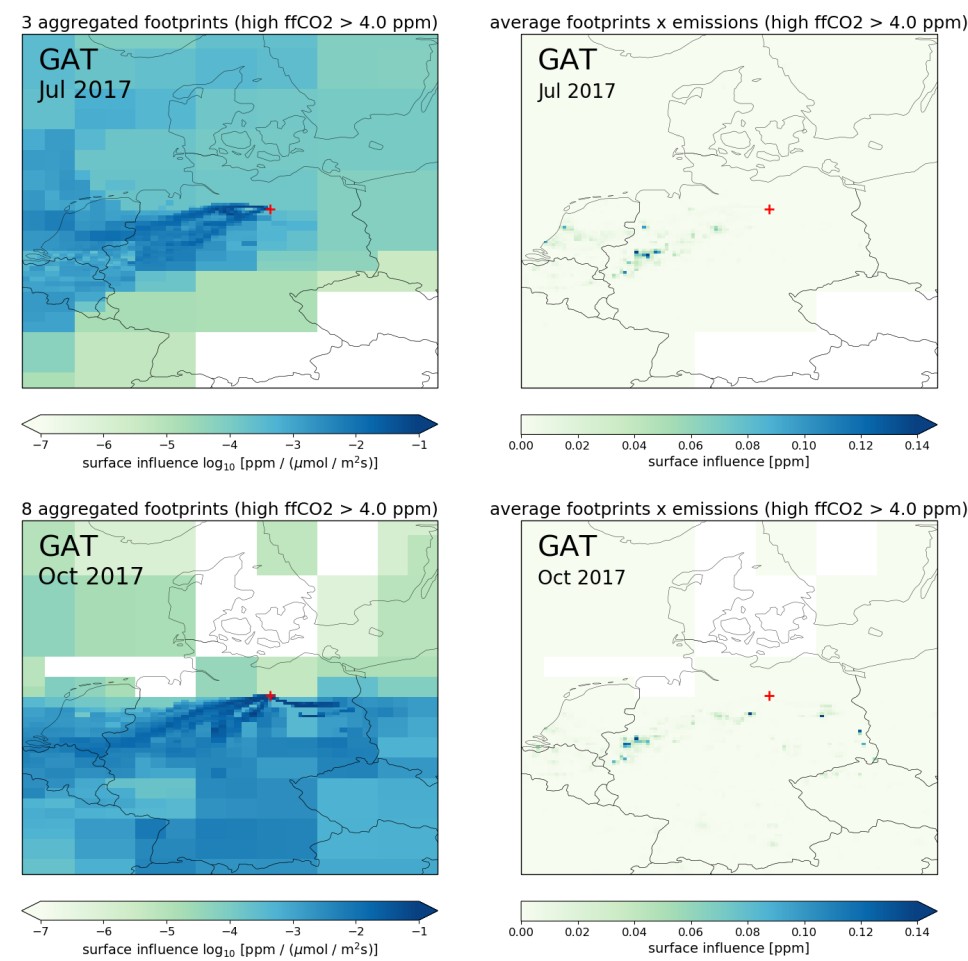


**Figure 10: Aggregated footprints with elevated ffCO$_2$ (left panels) and the corresponding surface influences (right panels) for Gartow in July 2017 (upper panels) and October 2017 (lower panels) based on the EDGARv 4.3 emission inventory. Note the logarithmic colour scale in the aggregated footprint maps.**



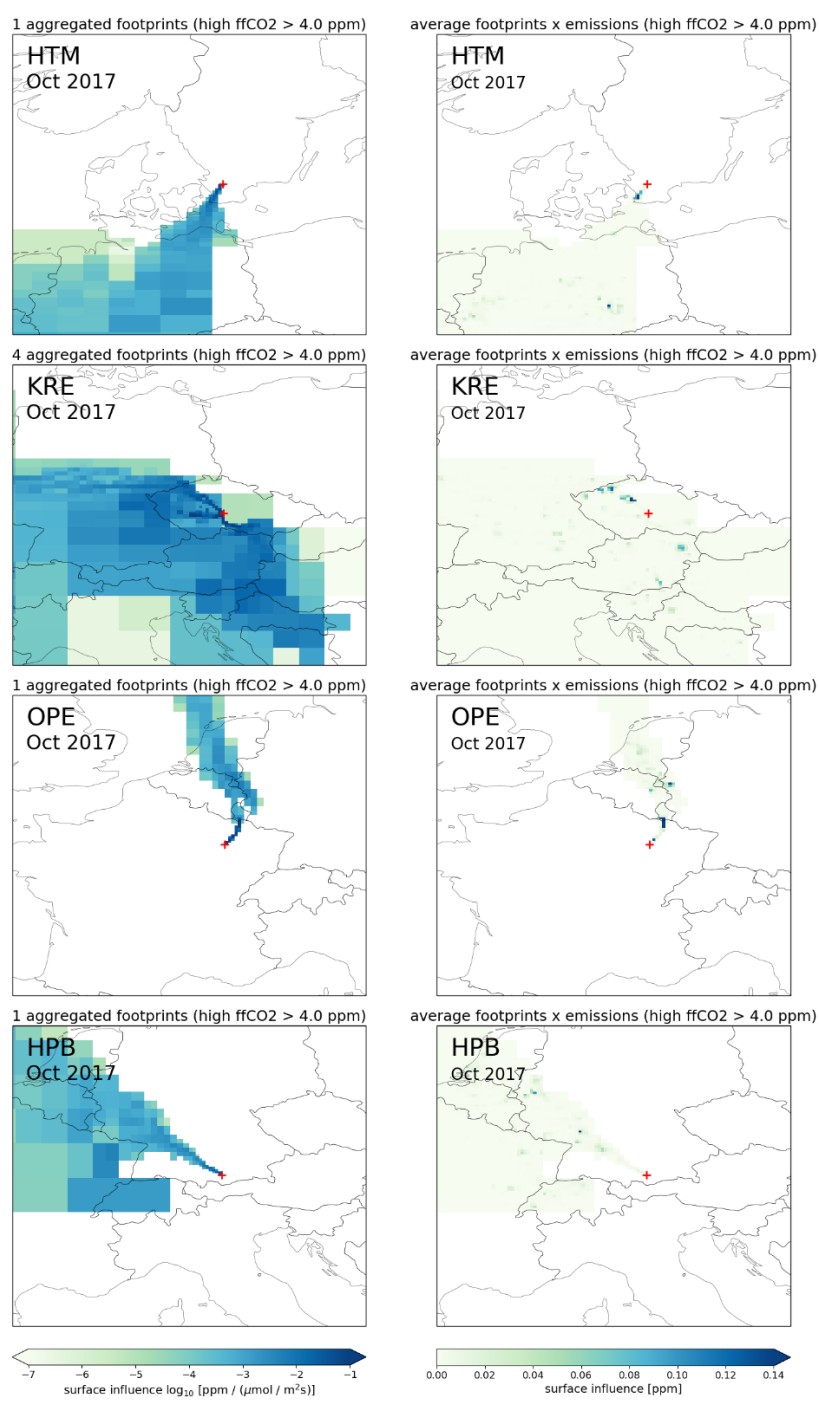


**Figure 11: Aggregated footprints with elevated ffCO$_2$ (left panels) and the corresponding surface influences (right panels) from top to bottom for Hyltemossa, Křešín, Observatoire Pérenne de l'Environnement and Hohenpeißenberg in October 2017.**
