# Peer review of "A dedicated flask sampling strategy developed for ICOS stations based on CO2 and CO measurements and STILT footprint modelling"

_Atmospheric Chemistry and Physics, 2020_

## Referee Comment (RC1) · Jocelyn Turnbull (Referee) · 15 Apr 2020

This paper outlines a strategy for collection of flask samples at the ICOS atmospheric stations. To my knowledge this is the most careful, detailed strategy available for utilising combined in situ and flask greenhouse gas measurements together. The paper discusses the reasons for using flask samples (quality checks on in situ measurements, measurement of additional species that are not or cannot be measured in situ). They discuss the strategy for when to collect flasks, and how to collect flasks, and the reasons that integrated sampling is useful. This is a very nice paper, and I recommend publication with minor revisions as detailed below.

[Figure]

Specific comments:

Edit for English grammar is needed throughout, but particularly in the introduction.

Abstract Line 33. The intent of this phrase is unclear.

Abstract Line 36. This is explained in the paper text, but it is unclear in the abstract why 4-5ppm is important.

Line 224. Six not sic

Line 246-248. This method differs somewhat from the Turnbull 2012 approach, which used a 15L integrating volume. In the ICOS case, the 3L flask itself is being used as the integrating volume. It would be helpful to see the weighting function that is used and a discussion of any impact the smaller integrating volume might have on the final integrated sample. This could be added as supplementary material.

Line 255. I'm guessing that this is because the flask itself is used as the integrating volume? If that is correct, please say so explicitly.

Line 271. Is there enough air for all these analyses in a single flask?

Line 395. This is a very nice demonstration of the utility of flask measurements in quality control. It is worth emphasizing in the conclusions that fast turnaround on the flask measurements AND speedy analysis of the results is needed to achieve outcomes like this.

Lines 468-469. I understand the motivation to measure 14C when the signals are large, but I wonder if this biased sampling methodology will be a headache in the end. If flask samples are biased towards high ffCO2 values, then the observed emission ratios might also be biased. For example, Turnbull et al 2011 measured 14C and CO from flasks collected in South Korea. The high ffCO2 values were associated with air masses coming from (nearby) Korea, which had low CO:ffCO2 ratios. Lower ffCO2 values were associated with air masses coming from China, which meant the ffCO2

signals were diluted. However, these Chinese samples had much higher CO:ffCO2 ratios. I could imagine a similar scenario in Europe. A secondary concern is that this data will presumably be shared across the wide ICOS network, and if these many and varied users are not aware of the deliberate sampling bias, there is a risk of misinterpretation of the results.

Line 479-480. References please.

Lines 525-527. I would argue that the 14C analysis would be most important in summer when the biospheric signals are larger and 14C is going to be even more critical for partitioning the fossil fuel and biospheric signals. This could be motivation to work towards higher precision 14C measurement capability.

Lines 546-547. In this case, will there be sufficient air remaining for a high precision 14C analysis?

Lines 553-555. Absolutely.

Lines 557-558. The NOAA tall tower network has an analogous flask program used for the same three goals. They collect flasks every 3 days (I think), but don't use integrated sampling. A point of difference would be that ICOS has taken a more thoughtful approach to design a sampling strategy to maximise the information from a minimum number of flasks.

---

## Referee Comment (RC2) · Auke van der Woude (Referee) · 23 Apr 2020

Comment on:
A dedicated flask sampling strategy developed for ICOS stations based on CO2 and CO measurements and STILT footprint modelling" by Ingeborg Levin et al.

By Auke van der Woude (auke.vanderwoude@wur.nl)

**General comments**

I think this is a very interesting and much needed research, helping the scientific measuring and modelling community. The paper discusses the potential uses of flask measurements for C14, measurements that are getting more and more important in climate research and proposes a sampling strategy for these measurements. The proposed sampling strategy for these flask measurements is justified with extensive research. To my knowledge, such a sampling strategy has not been created in such detail and with such justification before. Therefore, I recommend that this paper is published with minor revisions.

**Specific comments**

Especially in the introduction, the English is unclear and needs a revision.

Throughout the paper, the aims are discussed shortly thrice. One, more elaborate explanation would be better for me.

The introduction is unclear. In line 40-45, it is unclear why the measurements by Keeling are mentioned. Also a link between marine and terrestrial measurements is needed. The first two paragraphs should therefore be rewritten.

In Section 2.3, only STILT is mentioned. For stack emissions, STILT could introduce biases due to the representation on a grid. Therefore, a plume model might be needed for emissions from e.g. power plants. I would like to see a few sentences added that discuss this, either in Sect. 2.3 or in a discussion section.

In line 246, the '1/t filling approach' is mentioned. What is meant is unclear and should be explained or a reference should be added.

Line 290-295: Nighttime transport models are very erroneous. However, the integrated

flask samples are filled for two weeks. Does the limited capability of the nighttime transport not limit the usefulness of these integrated samples for modelers (as described in the aims), if they are also filled during night. (c.f. Line 294-295). I would like to see explained why the flasks are not only filled during well-mixed conditions.

Line 420 needs additional explanation: why are conditions with low ambient variability best suited to meet aim 1? It states that this is explained in the previous section, but this is unclear. Therefore, I would like to see this explicitly explained in the previous section.

In Section 4.2, only the results using flask measurements at 13.00 LT are shown. However, afternoon mixing conditions persist through about 16h in winter. Why are the flasks only filled at 13h? An analysis with the footprints for other times in the afternoon would add much information. I would like to see a paragraph (possibly including a figure), explaining the differences between sampling at 13h and other afternoon hours, and an explanation on why 13h is chosen.

In section 4.3, it is explained that mostly the winter C14 measurements are of interest. However, in winter time, the biosphere fluxes are small. Contrary, in summertime, the biosphere is very active and partitioning between biosphere and anthropogenic fluxes is very hard. C14 can help with this. Therefore, summertime flask samples are also very informative. I would like to see a sentence explaining, and possibly countering, this limitation.

**Technical corrections**

r. 37: How often can these events be expected?
r. 53: The mentioned process understanding needs more elaboration.
r. 50: These fluxes however, . . .
r. 81: ICOS flask sampling strategy might change in the future

r. 224: six
r. 350: in-situ
r. 369: suited to meet aim 1 (ongoing quality control at class 1 stations).
r. 420-421: Replace the could
r. 427: As expected, the regional coverage . . .
r. 493: In this summer month: What summer month?
r. 518: The mix of abbreviations and full names is confusing. A table with full name, location and abbreviation would help
r. 562: working successfully

*Figures and Tables:*

Table 1: This table is very full and therefore it is hard to find the needed information. A histogram might be more intuitive. This, however, is up to the authors to decide.

Figure 1: For better overview, it might be useful to indicate the sites in this figure.

Figure 2: The photograph of the sampler is superfluous. More explanation on the schematic is needed in the caption/text.

Figures 3 and 4 should be combined.

Figure 5: The title above the subfigures is very small and therefore unreadable.

Figure 6: The font and subplots are too small to read. It is also unclear what the main message of this figure is.

Figure 7: The x-axis could do with only one scale, also increasing the amount of space for the figures. Same goes for the y-axis

---

## Author Comment (AC2) · 30 Apr 2020

**Reply to the referee comments by Auke van der Woude**

We wish to thank Auke van der Woude for his careful review and helpful suggestions to improve the manuscript. Our answers and proposed changes are given below with the original comments printed in black and our replies in blue.

**General comments**

I think this is a very interesting and much needed research, helping the scientific measuring and modelling community. The paper discusses the potential uses of flask measurements for C14, measurements that are getting more and more important in climate research and proposes a sampling strategy for these measurements. The proposed sampling strategy for these flask measurements is justified with extensive research. To my knowledge, such a sampling strategy has not been created in such detail and with such justification before. Therefore, I recommend that this paper is published with minor revisions.

**Specific comments**

Especially in the introduction, the English is unclear and needs a revision.

We will check the English language (again) in the revised version of our manuscript and re-write the introduction.

Throughout the paper, the aims are discussed shortly thrice. One, more elaborate explanation would be better for me.

There are many different aspects concerning the sampling strategy in the ICOS network and we therefore feel that it may be helpful for the reader to remind him/her shortly of the specific aim before we discuss the particular aspect in detail. We can understand that this reviewer may like to have a more elaborate general introduction into the topic. For this we would, however, like to point him to the related literature, which we will extend.

Nevertheless, we will expand the first part of Sect. 3 in response to the reviewer's request.

The introduction is unclear. In line 40-45, it is unclear why the measurements by Keeling are mentioned. Also a link between marine and terrestrial measurements is needed. The first two paragraphs should therefore be rewritten.

We do not understand why we should NOT give credit here to the pioneering work of Dave Keeling !

We will re-write the introduction and make a better link between marine and terrestrial measurements.

In Section 2.3, only STILT is mentioned. For stack emissions, STILT could introduce biases due to the representation on a grid. Therefore, a plume model might be needed for emissions from e.g. power plants. I would like to see a few sentences added that discuss this, either in Sect. 2.3 or in a discussion section.

We mention here only STILT because this is the model we used for this study. We will add a short discussion on the problem of simulating correctly the influence of elevated point sources in Sec. 4.3.

In line 246, the '1/t filling approach' is mentioned. What is meant is unclear and should be explained or a reference should be added.

The 1/t filling approach is detailed in the reply to referee 1. We will give more information on this in the revised manuscript.

Line 290-295: Nighttime transport models are very erroneous. However, the integrated flask samples are filled for two weeks. Does the limited capability of the nighttime transport not limit the usefulness of these integrated samples for modelers (as described in the aims), if they are also filled during night. (c.f. Line 294-295). I would like to see explained why the flasks are not only filled during well-mixed conditions.

This is a very good point. Currently, we do the two-week integrated sampling day and night because this gives us a representative "continuous integrated" measure of the REAL MEAN ffCO2 concentration at the stations, with the prospect to observe potential changes in the future. This sampling scheme was our historical way to monitor 14CO2 at European stations and also globally, and we wanted to be compatible with these earlier measurements. It was also not our primary aim to collect these two-week integrated high volume samples in a way to best serve the modelling community, but rather for long-term monitoring purposes (although we are well aware of the fact that at sampling levels below 100 m height, the results would be biased towards night-time concentrations). We are still very hopeful that atmospheric transport models will improve in the near future and will then be able to "digest" also night-time observations. For these future times we will then have our observations ready to be used for validation of emissions changes. Still we always welcome the input from the experts, who tell us, e.g. via Observing System Simulation Experiments, how to optimise ICOS 14CO2 sampling strategies.

Line 420 needs additional explanation: why are conditions with low ambient variability best suited to meet aim 1? It states that this is explained in the previous section, but this is unclear. Therefore, I would like to see this explicitly explained in the previous section.

We would like to point the reviewer to Figs. 3 and 4 (left sides), which explicitly show that the agreement between 1-hour integrated flasks and in-situ measurements decreases with ambient variability. The reason for this is most probably that synchronization and weighting of flask filling and in-situ measurements (both collecting air through different intake lines and with different flow rates) is not perfect. These parameters are not so important when ambient variability is low; this was explained in section 4.1 (lines 379 ff).

In Section 4.2, only the results using flask measurements at 13.00 LT are shown. However, afternoon mixing conditions persist through about 16h in winter. Why are the flasks only filled at 13h? An analysis with the footprints for other times in the afternoon would add much information. I would like to see a paragraph (possibly including a figure), explaining the differences between sampling at 13h and other afternoon hours, and an explanation on why 13h is chosen.

It is correct that afternoon mixing conditions persist longer than one hour and flasks could potentially be sampled any other afternoon hour. Choosing 13 h LT for footprint analysis throughout the ms was just for consistency reasons, but it could have been any other afternoon hour. We do not agree that footprint analysis for other afternoon hours would add much information to this sensitivity study and in view of the aims of flask sampling at ICOS sites. Station PIs are free to choose a different afternoon hour or change the time every day. We will add a respective remark.

In section 4.3, it is explained that mostly the winter C14 measurements are of interest. However, in winter time, the biosphere fluxes are small. Contrary, in summertime, the biosphere is very active and partitioning between biosphere and anthropogenic fluxes is very hard. C14 can help with this. Therefore, summertime flask samples are also very informative. I would like to see a sentence explaining, and possibly countering, this limitation.

The limitation of summertime 14C-based ffCO2 estimates is simply current measurement precision and signal strength during summer. We will add a respective sentence in the conclusions. (See also reply to referee 1 who raised the same concern.)

**Technical corrections**

r. 37: How often can these events be expected?

We will add a range here.

r. 53: The mentioned process understanding needs more elaboration.

We will elaborate this in the revised manuscript (e.g. biospheric functioning)

r. 50: These fluxes however, . . .

Fine with us.

r. 81: ICOS flask sampling strategy might change in the future

Fine with us.

r. 224: six

Fine with us.

r. 350: in-situ

Fine with us.

r. 369: suited to meet aim 1 (ongoing quality control at class 1 stations).

Fine with us.

r. 420-421: Replace the could

We will change the sentence to "In the preceding section we showed that low ambient variability situations...."

r. 427: As expected, the regional coverage . . .

Fine with us.

r. 493: In this summer month: What summer month?

July (still related to Fig. 8)

r. 518: The mix of abbreviations and full names is confusing. A table with full name, location and abbreviation would help

We will replace the station acronyms by names throughout the text and in particular in line 518.

r. 562: working successfully

Fine with us.

Figures and Tables:

Table 1: This table is very full and therefore it is hard to find the needed information. A histogram might be more intuitive. This, however, is up to the authors to decide.

Good suggestion, we will try a histogram.

Figure 1: For better overview, it might be useful to indicate the sites in this figure.

Good suggestion, this should be possible to add (at least station abbreviations)

Figure 2: The photograph of the sampler is superfluous. More explanation on the schematic is needed in the caption/text.

We would like to keep the photograph (it does not need to be large but gives a good impression how the thing looks like), but add more information in the Figure caption for the schematics.

Figures 3 and 4 should be combined.

Yes, can easily be done.

Figure 5: The title above the subfigures is very small and therefore unreadable.

Will be combined to one title per row.

Figure 6: The font and subplots are too small to read. It is also unclear what the main message of this figure is.

We are sorry that the reviewer did not get the main message of this Figure as it presents the overall outcome of our representativeness analysis (aim 2): Aiming only for low variability situations for flask sampling creates serious biases in the data as shown in red in the lowest panel for each site. (Besides that we wanted to present the first years of continuous ICOS data that we are all proud of!) We will change the Figure and use the empty space in the lower right field for an enlarged legend (as it is the same in all 5 panels).

Figure 7: The x-axis could do with only one scale, also increasing the amount of space for the figures. Same goes for the y-axis

We prefer not to change that Figure because individual panels are easier to read with individual axes. Space is not a problem here (contrary to Fig. 6).

---

## Author Comment (AC1)

**Reply to the referee comments by Jocelyn Turnbull**

We wish to thank Jocelyn Turnbull for her very thoughtful comments and suggestions to improve the manuscript. Our answers and proposed changes are given below with the original comments printed in black and our replies in blue.

This paper outlines a strategy for collection of flask samples at the ICOS atmospheric stations. To my knowledge this is the most careful, detailed strategy available for utilizing combined in situ and flask greenhouse gas measurements together. The paper discusses the reasons for using flask samples (quality checks on in situ measurements, measurement of additional species that are not or cannot be measured in situ). They discuss the strategy for when to collect flasks, and how to collect flasks, and the reasons that integrated sampling is useful. This is a very nice paper, and I recommend publication with minor revisions as detailed below.

Specific comments:

Edit for English grammar is needed throughout, but particularly in the introduction.

We will check English grammar (again) in the revised version of our manuscript.

Abstract Line 33. The intent of this phrase is unclear. *(Phrase: "In order to have a maximum chance to also sample ffCO2 emission areas, additional flasks need to be collected on all other days in the afternoon."*

The likelihood to sample flasks that are significantly (by about 6 ‰ (2 sigma)) "contaminated" by ffCO2 is low, as ICOS stations are located far away from anthropogenic emissions. In order to "catch" ffCO2 events, which can be used in dedicated inversions we thus have to sample flasks frequently and check afterwards, using the continuous CO measurements at the stations, which flasks should indeed be kept for 14CO2 analysis.

We will add text to make this issue more clear already in the abstract.

Abstract Line 36. This is explained in the paper text, but it is unclear in the abstract why 4-5ppm is important.

We will add the explanation i.e. including after 4-5ppm ", *that allows ffCO2 estimates with an uncertainty below 30%"*

Line 224. Six not sic

Thanks

Line 246-248. This method differs somewhat from the Turnbull 2012 approach, which used a 15L integrating volume. In the ICOS case, the 3L flask itself is being used as the integrating volume. It would be helpful to see the weighting function that is used and a discussion of any impact the smaller integrating volume might have on the final integrated sample. This could be added as supplementary material.

An almost constant weighting of the sample concentration over the one-hour sampling time is achieved by the temporal modulation of the sample flow f(t) [SLPM] passing a flask, which acts at the same time as mixing volume V given in liter STP.

The flow rate f is changed according to $f(t) = V/(t - t_0)$ over time t. Since the flow rate at the start time $t_0$ in a 1/time function would be infinite, which is not possible in reality, a 30 minute flushing phase at maximum flow rate precedes the averaging phase to ensure the complete air exchange in the flask.

The concentration $c_F(t)$ in the flask is determined by the ambient air concentration $c_A(t)$ and can be described as time series using sufficiently small time steps $\Delta t$:

$$c_F(t + \Delta t) = \frac{c_F(t) \cdot (V - f(t) \cdot \Delta t) + c_A(t) \cdot (f(t) \cdot \Delta t)}{V} \approx c_F(t) + c_A(t) \cdot \frac{f(t) \cdot \Delta t}{V}$$

The resulting weight of the ambient air concentration $w_{c_A}$ at time step $t_n$ in the flask depends on two factors:

$$w_{c_A}(t_n) \sim c_A(t_n) \cdot \frac{f(t_n) \cdot \Delta t}{V} \cdot \prod_{i=n+1}^{E} \left(1 - \frac{f(t_i) \cdot \Delta t}{V}\right)$$

the weight at the moment when the ambient air portion enters the flask and a weight reduction factor caused by dilution with sampled air entering the flask at later times. The reduction is calculated by multiplication of the respective dilution steps from $t_n$ to the sampling end time $t_E$.

Figure 1 shows an example flow and weighting function over time from a sampling event. The weight functions are calculated from the respective flow controller and pressure sensor measurements for each single sampling event.

[Figure]

Fig. 1: Flow-rate and respective weighting function from a sampling event using a 3 liter flask.

For the results originally presented in Figs. 3 and 4 of the manuscript we used a slightly different weighting function (i.e. according to Turnbull et al., 2012) than described above. Re-evaluation of the comparison data for Heidelberg with the algorithm shown above did not significantly change the outcome (slightly improved it). The more correct evaluation method described here will be applied for all Heidelberg and Hohenpeißenberg comparison data and presented in the revised version of the manuscript.

Unfortunately, we are not in the position to discuss the impact a small integrating volume may have as we did not make any respective experiments. Up until now we are confident that it has no impact at all.

A comprehensive description of the suggested universal 1/t approach is currently in preparation and will be published together with a more detailed description of the flask sampler itself and its potential applications (Eritt et al., in preparation). This will also concern description of the possibility of more elaborated sampling strategies, such as triggering through trajectory forecasts. In the current manuscript we will thus limit description of sample integration to a minimum and point the reader to a more elaborate upcoming paper.

Line 255. I'm guessing that this is because the flask itself is used as the integrating volume? If that is correct, please say so explicitly.

Not exactly: In order to sample two flasks exactly in parallel, one needs two flow controllers (independent of the fact that the flow rate through the flask is changed during integrated sampling (Fig. 1) or not). This is mainly due to slightly different flow resistances of tubing and valves when flushing the 24 different flasks, as was mentioned in lines 255 ff of the manuscript.

Line 271. Is there enough air for all these analyses in a single flask?

The 3 Liter flasks are pressurized to 1.7 bar abs, thus providing a total sample volume of 5100 ml STP. The following volumes (STP) are consumed by the different analyses in the FCL:

GHG (duplicate analysis): 120 ml,
Stable isotope ratios of $CO_2$: 600 ml,

O2/N2: 150ml.

The remaining sample volume for the 14C AMS analysis is, thus, about 4100 ml. In practice, the volumes analysed for 14CO2 vary between 1000 ml and 5500 ml with a median of 4200 ml. This variation is caused by the use of smaller flask sizes and by the fact that not for all flasks all analyses were yet conducted before CO2 extraction for AMS analysis. The AMS error for a single sample is composed of the counting error and the uncertainty of the calibration, which is determined using oxalic acid standards and blanks according to Wacker et al. (2010). For sample volumes larger than 2500 ml our mean AMS error is (1.9±0.4) ‰. For sample volumes smaller than 2500 ml the AMS error increases to about 3‰ for sample volumes of 1500 ml only.

In summary: Yes, there is enough air in the flasks for precise analysis of all currently envisaged components.

Line 395. This is a very nice demonstration of the utility of flask measurements in quality control. It is worth emphasizing in the conclusions that fast turnaround on the flask measurements AND speedy analysis of the results is needed to achieve outcomes like this.

Yes, Jocelyn, you are right. We will mention this in the conclusions !

Lines 468-469. I understand the motivation to measure 14C when the signals are large, but I wonder if this biased sampling methodology will be a headache in the end. If flask samples are biased towards high ffCO2 values, then the observed emission ratios might also be biased. For example, Turnbull et al 2011 measured 14C and CO from flasks collected in South Korea. The high ffCO2 values were associated with air masses coming from (nearby) Korea, which had low CO:ffCO2 ratios. Lower ffCO2 values were associated with air masses coming from China, which meant the ffCO2 signals were diluted. However, these Chinese samples had much higher CO:ffCO2 ratios. I could imagine a similar scenario in Europe. A secondary concern is that this data will presumably be shared across the wide ICOS network, and if these many and varied users are not aware of the deliberate sampling bias, there is a risk of misinterpretation of the results.

This is a valid point and totally correct: We have indeed to make sure that the flask sample results are used in a proper way. It is, however, not our aim to determine CO/ffCO2 ratios and then use those for e.g. estimating continuous ffCO2 concentrations as in Vogel et al. (2010). Here CO is only used as indicator for potentially high ffCO2 concentrations at the time of flask sampling. We expect that this dedicated sampling has advantages when the results are used in regional inversions to estimate the ffCO2 component. If we always measure 14C-based ffCO2 signals at the detection limit, this increases the uncertainty of the results and we may not meet our primary aim 3 to determine the ffCO2 component. As presented in Fig. 7, we collect, in addition to the flasks, two-week integrated 14CO2 samples, which sample all footprints and will thus provide representative results.

We will add a corresponding note in the revised manuscript.

Line 479-480. References please.

We will add the following references:

Gamnitzer et al., 2006; Turnbull et al., 2006 ; Levin and Karstens, 2007 ; Vogel et al., 2010 ; Turnbull et al., 2011.

Lines 525-527. I would argue that the 14C analysis would be most important in summer when the biospheric signals are larger and 14C is going to be even more critical for partitioning the fossil fuel and biospheric signals. This could be motivation to work towards higher precision 14C measurement capability.

Again Jocelyn Turnbull is very right and we are all working hard towards this aim (summer ffCO2 estimates and better precision in 14C analysis). It is clear and in fact it was on purpose NOT to design the ICOS network for monitoring ffCO2 emissions but to monitor European ecosystem fluxes and their changes as a first priority. We are thus happy to see that our original ICOS network design was successful in measuring only small fossil fuel CO2 concentrations. Our first step and currently realistic aim with flask sampling is trying to constrain winter time ffCO2 (emissions). The ongoing two-week integrated 14CO2 sampling at the ICOS stations provides whole-year representative data, also covering the vegetation period.

Lines 546-547. In this case, will there be sufficient air remaining for a high precision 14C analysis?

Yes, there is enough air in the flasks for precise analysis of all currently envisaged components (see our comment above).

Lines 553-555. Absolutely.

This is in fact our main concern and we hope that the modelling community (and funding agencies) will be able to support this aim soon.

Lines 557-558. The NOAA tall tower network has an analogous flask program used for the same three goals. They collect flasks every 3 days (I think), but don't use integrated sampling. A point of difference would be that ICOS has taken a more thoughtful approach to design a sampling strategy to maximise the information from a minimum number of flasks.

We will take up this comment in our conclusions.

References:

Gamnitzer, U., Karstens, U., Kromer, B., Neubert, R. E M., Meijer, H. A. J., Schroeder, H. and Levin, I.: Carbon monoxide: A quantitative tracer for fossil fuel $CO_2$? J. Geophys. Res., 111, D22302, doi:10.1029/2005JD006966, 2006.

Levin, I. and Karstens, U.: Inferring high-resolution fossil fuel CO2 records at continental sites from combined $(CO_2)$-C-14 and CO observations, Tellus B., 59, 245–250, 2007.

Turnbull, J. C., Miller, J. B., Lehman, S. J., Tans, P. P., Sparks, R. J. and Southon, J.: Comparison of $^{14}CO_2$, CO, and $SF_6$ as tracers for recently added fossil fuel $CO_2$ in the atmosphere and implications for biological $CO_2$ exchange, Geophys. Res. Lett., 33: L01817, doi: 10.1029/2005GL024213, 2006.

Turnbull, J. C., Tans, P.P., Lehman, S. J., Baker, D., Conway, T. J., Chung, Y. S., Gregg, J., Miller, J. B., Southon, J. R.,and Zhou, L.-X.: Atmospheric observations of carbon monoxide and fossil fuel CO2emissions from East Asia, J. Geophys.Res.,116, D24306, doi:10.1029/2011JD016691, 2011.

Vogel, F.R., Hammer, S., Steinhof, A., Kromer, B. and Levin, I.: Implication of weekly and diurnal $^{14}C$ calibration on hourly estimates of CO-based fossil fuel $CO_2$ at a moderately polluted site in south-western Germany, Tellus, 62B, 512-520, doi: 10.1111/j.1600-0889.2010.99466.x, 2010

Wacker, L., Christl, M., & Synal, H. A.: Bats: a new tool for AMS data reduction. Nuclear Instruments and Methods in Physics Research Section B: Beam Interactions with Materials and Atoms, 268(7-8), 976-979, 2010.